# Robust Gaussian Process Regression with Huber Likelihood

## Abstract

Outliers in both covariates and output responses pose significant challenges for Gaussian Process (GP) regression models. We present a novel GP regression approach that effectively integrates the Huber likelihood into the GP framework—without introducing additional parameters to infer. Specifically, we model the likelihood of observed outputs using the Huber probability distribution: this reduces deviations caused by output outliers. For covariate outliers, we introduce a projection pursuit weights—attenuating their influence on the model. To address the analytically intractable, yet unimodal, posterior distribution, we employ Laplace approximation and Gibbs sampling within a Markov Chain Monte Carlo (MCMC) framework. We simplify Gibbs sampling by expressing the likelihood associated with outlying points as normally distributed through the scale mixture representation of the Laplace distribution. This work is particularly important in the field of transmission spectroscopy—where noisy measurements are often neglected in the estimation of planet-to-star radius ratios. We demonstrate the robustness and effectiveness of our method through extensive experiments on synthetic and real-world datasets.

## 1 Introduction

Bayesian inference which is based on Gaussian likelihood is known to be sensitive to extreme observations and gross errors, called outliers. The estimation of parameters in Gaussian processes (GPs) is affected in non-Gaussian error settings as the predictive uncertainty assigns equal confidence to the measurements, regardless of whether they are outliers or not. We illustrate this problem in a numerical example. Let us consider a 2-d sinc function $y(x) = \text{sinc}(x) + e$, where $x = \sqrt{(x_1^2 + x_2^2)}$ with an additive error that follows the Student's t-distribution with 2 degrees of freedom $e \sim$ Student's-t(2). We add additional large outliers $y^l$ with magnitude close to 0.8. Figure 1(b) shows the predicted values at test points $x = [-16, 16]$, obtained from standard GP. We observe that the mean fit deviate largely from the true values of the sinc function. This issue is amplified by multiple outliers masking one another in the multivariate regression residuals.

Existing studies addressing the outlier problem in GP regression use various approaches to define the likelihood. Two common strategies are: (1) using a mixture of two normal distributions or (2) employing heavy-tailed distributions. Most of these methods assume the error distribution is known a priori—a condition that is often unrealistic in practical applications. Moreover, their robustness is questionable when faced with extreme observations that do not correspond to the non-normal distribution their heavy tailed likelihood is specified to capture. These models typically struggle to handle both general noise patterns and large errors, often attempting to fit extreme values. We show this shortcoming in Figure 1(a) with the sinc function data for the GP with the Student's t-likelihood and employing the MCMC integration approximation method. We notice that the model overfits to the large outliers $y^l$ as it accounts for the large errors with Student's-t likelihood.

In this paper, we propose a new way of handling extreme outliers in covariate space and output responses that models the likelihood of the observed data using Huber density function. We significantly enhance downweighting of the outliers compared to the earlier work by Altamirano et al. (2024), which was limited to handling outliers only in the output responses, added hyperparameters $(\beta, c)$, did not support maximum likelihood estimation of hyperparameters, and was incapable of addressing outliers occurring simultaneously in the covariate and output space.

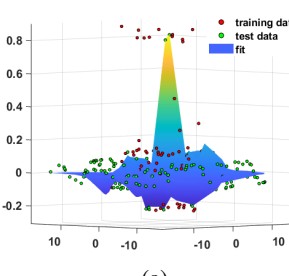 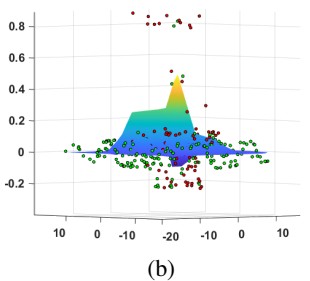

(a)            (b)

Figure 1: Predictions of the $\text{sinc}(x)$ where the added errors are $i.i.d$ according to the Student-t distribution with 2 degrees of freedom, that is, $e \sim$ Student's t(2), obtained from (a) Student's t-likelihood with the MCMC integration and (b) GP with MAP estimates.

## 2 RELATED WORK

Goldberg et al. Goldberg et al. (1997) introduced a dual-model Gaussian process framework to account for covariate-dependent noise. The first Gaussian process model governs the output process $y$, while the second Gaussian process governs the noise process. West (1984) investigated heavy-tailed error distributions that are constructed as scale mixtures of normal distributions, which are also used for specifying a priori distribution based on the earlier ideas suggested by De Finetti (1961); Ramsey & Novick (1980). By doing so, the prior distribution discounts any observations highlighting inconsistency between likelihood and prior. Along the same line, Desgagné & Gagnon (2019) assumed a super heavy-tailed error distribution dependent on an explanatory variable to make the estimation of the population mean and ratios robust to outliers. Kuss (2006) extended a mixture of two normal distributions, one to model small errors in regular observations and a second one to model large errors in outlying observations. However, Naish-Guzman & Holden (2007) questioned the adequacy of the two-model approach. They proposed instead twin GP that allow us to choose between the distribution of the regular observations and that of the outliers. Kuss (2006) suggested a GP with a Laplace likelihood model that utilizes a scale mixture representation of Laplace noise distribution where the variance follows an exponential distribution. Vanhatalo et al. (2009) proposed a GP model based on the Student's t-likelihood function, where the noise is modeled as a scale mixture of Gaussian distributions. Unfortunately with the non-Gaussian likelihood, the Bayesian inference becomes analytically intractable. Consequently, various advanced approximation methods were proposed Kuss (2006); Vanhatalo et al. (2009); Jylänki et al. (2011); Ranjan et al. (2016); Daemi et al. (2019) to overcome the convergence failure of the classical approximation methods such as expectation propagation Minka (2013), Markov Chain Monte Carlo Neal (1997), variational Bayes Ghahramani & Beal (2000), and Laplace approximation Williams (1996). More recently, Li et al. (2021); Andrade & Takeda (2023) presented a robust variants of GPs for datasets with substantial contamination removing the outlier data based on trimming parameters in iterative manner.

In GP regression models with Student's t-likelihoods Kuss (2006), a scale-mixture representation of the Student's t-distribution is utilized. A variational approximation is devised presuming the Gaussian likelihood whose individual variances are Gamma distributed. Combined with the Kullback-Leibler divergence, $\text{KL}(q||p)$, between the true posterior, $p$, and the approximation, $q$, an expectation maximization (EM)-type algorithm is implemented. As for the models with Laplace likelihoods, the scale mixture model yields a unimodal posterior enabling the implementation of the EP approximation and the MCMC sampling. Here, a Laplace approximation is inappropriate because the discontinuous derivatives of the Laplace likelihood at zero may cause the Hessian matrix to be undefined.

## 3 CONTRIBUTIONS

Altamirano et al. Altamirano et al. (2024) proposed a robust Gaussian Process (GP) regression method that uses generalized Bayesian inference: an approach designed to preserve computational conjugacy at the expense of maximum likelihood estimation for the GP hyperparameters. Their method handles outliers in the output responses through weighting mechanism $J$ in the noise term: $\sigma^2 J_{ii} = \sigma^2 \left(1 + r_i^2/c^2\right)$, where $r_i$ is the residual associated with $i^{\text{th}}$ data point $r_i = y_i - m(\boldsymbol{x}_i)$ and $c$ is the threshold parameter. However, this approach has a critical limitation: it fails to account for outliers in the output response, $y_i^{(c)}$, when they occur alongside outliers in the covariate dimensions,

$\boldsymbol{x}_i = [x_1^{(c)}, x_2^{(c)}, \ldots, x_d]$. As a result, the method's accuracy diminishes since their downweighting only targets outliers based on the discrepancy $y_i - m(\boldsymbol{x}_i)$, neglecting the impact of covariate outliers.

Our approach begins by transforming the contaminated data points, $\{y_i^{(c)}, \boldsymbol{x}_i^{(c)}\}_{i=1}^{n_c}$, into a more reliable dataset using projection pursuit weights $w(\boldsymbol{x})$. These weights are then applied to scale the residuals $r$, ensuring that the influence of an outlier is adjusted based on the presence of extreme covariate outliers $\boldsymbol{x}_i^{(c)}$. To further handle extreme outliers in output response $y_i^{(c)}$, we employ a Huber density function—derived from the exponential of the Huber loss- giving robust L1 norm treatment for the residuals having over-limit magnitude.

Our approach does not introduce Huber likelihood specific parameters in the posterior inference, avoiding the need for additional model-specific tuning, and is effective under non-Gaussian noise conditions. Notably, when extreme outliers are detected in the covariate dimensions $\boldsymbol{x}_i^l$, the model selectively retains the corresponding output $y_i^l$ if it enhances the regression fit.

## 4    THE MODEL

Let us consider a regression setting $y_i = f(\boldsymbol{x}_i) + \epsilon_i$, where $\epsilon_i \sim \mathcal{N}(0, \sigma^2)$ is a homoscedastic i.i.d. random variable with constant variance. In GP models, the systematic dependency between the covariates $\boldsymbol{x}$ and output vector $y \in \mathbb{R}$ is given by a latent function, $f(\boldsymbol{x}) : \mathbb{R}^d \to \mathbb{R}$. In a truly non-parametric sense, the latent vector function at $n$ covariates, $\mathbf{f} = [f(\mathbf{x}_1), \ldots, f(\boldsymbol{x}_n)]^\top$, is assumed to have a priori probability distribution. This distribution is a joint multivariate normal distribution with zero mean vector and covariance matrix, $\mathbf{K}$, that is,

$$\mathbf{f}|\mathbf{X}, \boldsymbol{\theta} \sim \mathcal{N}(\mathbf{f}|\mathbf{0}, \mathbf{K}). \tag{1}$$

The covariance matrix, $\mathbf{K}$, is a positive semi-definite matrix that captures residual spatial association with elements $K_{i,j} = k(\boldsymbol{x}_i, \boldsymbol{x}_j)$, $i, j = 1, \ldots, n$. The function $k(\cdot, \cdot)$, chosen from a parametric kernel family such as the Gaussian or the Matérn kernel, is characterized by hyperparameters denoted by $\boldsymbol{\theta}$. The likelihood of the data is expressed as $\mathbf{y}|\mathbf{f}, \sigma \sim \mathcal{N}(\mathbf{y}|\mathbf{f}, \boldsymbol{\Sigma})$, and the resulting posterior distribution on $\mathbf{f}$ as where $\boldsymbol{\Sigma} = \text{diag}(\sigma_1^2, \ldots, \sigma_n^2)$.

Next, we develop three aspects of the proposed GP-Huber model: Huber likelihood, projection pursuit weights, and the resulting unimodal posterior distribution. Following that, we discuss the hyperparametric settings of the GP-Huber.

### 4.1    HUBER LIKELIHOOD

We propose to use the Huber density function based on the Huber loss proposed by Huber (1992) to model the likelihood of the observed data. The Huber loss function $\rho(\cdot)$ is a truncated mixture of two commonly used loss functions: squared loss, $l(r) = r^2$ for residuals below threshold $b$, and absolute loss, $l(r) = |r|$ for residuals $r_i = y_i - f(\mathbf{x}_i)$ below threshold $b$, given by

$$\rho(r) = \begin{cases} \frac{1}{2}r^2, & \text{if } |r| \leq b \\ b|r| - \frac{1}{2}b^2. & \text{otherwise} \end{cases} \tag{2}$$

Huber (1992) considered the contamination model $(1 - \varepsilon)G(r) + \varepsilon H(r)$, where $G(r)$ is the Gaussian cumulative density function and $H(r)$ is the unknown cumulative density function. The associated least favorable Huber density function with a fraction of contamination $\varepsilon$ is defined as

$$p_H(\mathbf{y}|\mathbf{f}, \phi) = \prod_{i=1}^{n} \frac{1 - \varepsilon}{\sqrt{2\pi}\sigma} \exp\left(-\rho(r_i)\right). \tag{3}$$

The parameter $\varepsilon$, symbolizing the fraction of the dataset presumed to deviate from the underlying model, can be computed utilizing the minimum covariance determinant estimator Hubert & Debruyne (2010). The threshold $b$ is selected to protect estimation of the model parameters and hyperparameters against the fraction of contamination $\varepsilon$.

### 4.2    PROJECTION PURSUIT WEIGHTING

The idea is to scale the residual $r_i$ associated with the $i^{\text{th}}$ data point with projection pursuit weight $w(\boldsymbol{x}_i)$ based on robust variant of Mahalanobis distances, called projection statistics $\text{PS}(\boldsymbol{x}_i) : \mathbb{R}^d \to$

$\mathbb{R}^d$. This scaling highlights the impact of outliers in single or multiple dimensions masking each other in the covariate space. Residual larger than the threshold $b$ gets robust $L1$ norm treatment, while those smaller than $b$ are treated with an efficient $L2$ norm within the Huber loss $\rho(r)$.

We obtain standardized the residual $r_{S_i} = r_i/(w_i\sigma s)$ by scaling $r_i$ by its corresponding projection pursuit weight $w_i$ and using a scaling factor $s = b_d \; \text{med}|\boldsymbol{r}|$, where $b_d = 1 + 5/(n-d)$ is the dimensionality correction factor. When the error distribution is unknown, $s$ accounts for its spread parameter. The projection pursuit weights $\boldsymbol{w}$ limit the influence of outliers simultaneously arising in multiple covariate dimensions at multiple locations on the loss function, are based on projection statistics $\text{PS}_i$, calculated as

$$w_i = \begin{cases} 1, & \text{for } \text{PS}_i^2 \le c_i, \\ \frac{c_i}{\text{PS}_i^2}, & \text{for } \text{PS}_i^2 > c_i. \end{cases} \tag{4}$$

The projection statistics (Stahel, 1981; Donoho, 1982) are a robust version of Mahalanobis distances based on the median absolute distance from the median. Formally defined as the maxima of the standardized projection distances obtained by projecting the point cloud in the directions that originate from the co-ordinate wise median and that pass through each of the data points, $\boldsymbol{x}_i$ (Mili et al., 1996). They're easy to calculate:

$$\text{PS}_i = \max_{||\boldsymbol{u}_j||=1} \frac{|\boldsymbol{x}_i^T\boldsymbol{u}_j - \underset{k}{\text{median}}(\boldsymbol{x}_k^T\boldsymbol{u}_j)|}{1.4826 \, \underset{i}{\text{median}} \, |\boldsymbol{x}_i^T\boldsymbol{u}_j - \underset{k}{\text{median}}(\boldsymbol{x}_k^T\boldsymbol{u}_j)|}, \tag{5}$$

where $\boldsymbol{u}_j = \frac{\boldsymbol{x}_j - \mathbf{M}}{||\boldsymbol{x}_j - \mathbf{M}||}$; $j, k = 1, \ldots, n$. The co-ordinate wise median $\mathbf{M}$ is given by $\mathbf{M} = \{\underset{j=1,\ldots,n}{\text{med}} \, \boldsymbol{x}_{j1}, \ldots, \underset{j=1,\ldots,n}{\text{med}} \, \boldsymbol{x}_{jd}\}$. The projection statistics attain the maximum breakdown point given by $[(n-d-1)/2]/n$ (Maronna & Yohai, 1995).

Stahel et al. (1991) and Mili et al. (1996) showed that, when $n > 5d$, the squared projection statistics $\text{PS}_i^2$ roughly follow a $\chi^2$ distribution with a degree of freedom equal to the number of non-zero elements $\nu_i$ in the row vector of the associated regressor, $\mathbf{x}_i$, i.e., $\text{PS}_i^2 \sim \chi_{\nu_i}^2$. However, when $n \le 5d$, it is the PS that roughly follow a $\chi^2$ distribution, that is, $\text{PS}_i \sim \chi_{\nu_i}^2$. Consequently, the threshold $c_i$ is chosen as the 97.5 percentile of the chi-square distribution with $\nu_i$ degrees of freedom while defining weights in equation 4.

Throughout the inference process (as detailed in Section 5), we use standardized residuals $r_{S_i}$ within the Huber likelihood.

$$p_H(\mathbf{y}|\mathbf{f}, \phi) = \prod_{i=1}^{n} \frac{1-\varepsilon}{\sqrt{2\pi\sigma}} \exp\left(-\rho(r_{S_i})\right). \tag{6}$$

### 4.3 GP-HUBER POSTERIOR

The posterior distribution resulting from our model, which incorporates a non-conjugate prior, is given as:

$$p(\mathbf{f}|\mathcal{D}, \boldsymbol{\theta}, \sigma) = \frac{p_G(\mathbf{f}|\mathbf{0}, \mathbf{K})}{p(\mathcal{D}|\boldsymbol{\theta}, \sigma)} p_H(\mathbf{y}|\mathbf{f}, \sigma), \tag{7}$$

where where $p_G(\mathbf{f}|\mathbf{0}, \mathbf{K})$ is the Gaussian prior $\mathcal{N}(\mathbf{f}|\mathbf{0}, \mathbf{K})$ and $p_H(\mathbf{y}|\mathbf{f}, \sigma)$ is the likelihood modeled using the Huber density. This formulation leads to a posterior that does not have a closed-form expression due to the non-conjugate nature of the Huber likelihood.

The marginal likelihood (or evidence) of the data, which plays a crucial role in model selection and hyperparameter optimization, is expressed as:

$$p(\mathcal{D}|\sigma, \boldsymbol{\theta}) = \int p_G(\mathbf{f}|\mathbf{0}, \mathbf{K}) p_H(\mathbf{y}|\mathbf{f}, \sigma) d\mathbf{f}. \tag{8}$$

**Theorem 1.** *The GP-Huber posterior distribution $p(\mathbf{f}|\mathcal{D}, \boldsymbol{\theta}, \sigma)$ is unimodal.*

The proof can be found in Appendix A.1. This theorem indicates that despite the non-Gaussian and potentially complex nature of the Huber likelihood, the posterior retains a single peak, simplifying both inference and hyperparameter optimization.

We can set the threshold $b = 1.5$ to achieve high efficiency at the Gaussian distribution (see appendix A.2). This would make our model robust to $10\%$ outliers (since fraction of contamination is $\varepsilon = 0.1$). Note that, in the context of our work, "efficiency" refers to the estimator's ability to achieve low variance when the noise follows a Gaussian distribution. Specifically, a highly efficient estimator can make the best use of data that is predominantly Gaussian, leading to more accurate parameter estimation. The contamination fraction $\varepsilon$ defines the model's tolerance to deviations from the Gaussian assumption, allowing it to handle a proportion of outlier points without being overly influenced by them. The parameter $b$ controls the threshold for identifying outliers and thus influences the transition between $L2$ and $L1$ norm treatment. By setting $b = 0.45$, we get $\varepsilon = 0.45$ for heavy-tailed and Gaussian error distributions, we aim to accommodate up to $45\%$ outliers while maintaining reasonable efficiency. The only hyperparameter of the likelihood function requiring estimation is $\phi = \sigma^2$. Thus, the incorporation of projection pursuit weighting and the Huber likelihood does not introduce any extra hyperparameters.

## 5 APPROXIMATE BAYESIAN INFERENCE

By retaining the optimization-friendly properties of convex problems ensured by to unimodality (see Theorem 1), our method enables the use of the Laplace approximation (Tierney & Kadane, 1986) for the posterior. To facilitate predictions $f^*$, we develop Gibbs sampling and Laplace's method. The key requirement for the latter is the continuity of the Huber density function, ensuring that its derivatives exist for all $r_S$ in the interval $(-\infty, \infty)$. In Gibbs sampling, the joint posterior distribution $p(\mathbf{f}, \boldsymbol{\theta}, \sigma^2)$ can be simplified using the scale mixture model of the Laplace distribution for data points with residuals $r \geq b$: this representation expresses the likelihood of these points as a normal distribution—making the sampling process more efficient.

### 5.1 GIBBS SAMPLING

The Huber density function is a mixture of a truncated normal and a Laplace density function for an absolute standardized residual respectively lying within and outside the threshold $b$. This yields

$$p_H(y|f, \boldsymbol{\sigma}) = \begin{cases} \frac{C_1}{\sqrt{2\pi} w_i \sigma_g s} \exp\left(-\frac{r_i^2}{2 w_i^2 \sigma_g^2 s^2}\right) & |r_{S_i}| \leq b, \\ \frac{C_2}{2 w_i a s} \exp\left(-\frac{b|r_i|}{w_i a s}\right) & |r_{S_i}| > b, \end{cases} \tag{9}$$

where $C_1$ and $C_2$ are the constants respectively, defined as $C_1 = 1 - \varepsilon$ and $C_2 = \sqrt{\frac{\pi}{2}} \exp(b^2/2)$. The Laplace distribution $p_L(y_i|f(\mathbf{x}_i), a)$ with location parameter $a$ can be represented as a scale mixture of normal distributions $\mathcal{N}(y_i|f(\mathbf{x}_i), \sigma_i^2)$ where $\sigma_i^2$ follows an exponential distribution $p_E(\sigma_i^2|\beta)$ Andrews & Mallows (1974) and $i = 1, \ldots, n_l$ are the indices of the points associated with the standardized residuals larger than the threshold $b$ hereafter referred to as outlying points. Formally, we have

$$p_L(y_i|f(\mathbf{x}_i), a) = \int p_G(y_i|f(\mathbf{x}_i), \sigma_i^2) p_E(\sigma_i^2|\beta) d\sigma_i^2. \tag{10}$$

Using this property, we represent the individual standard deviations corresponding to $n_l$ outlying training points as $\{\sigma_{l_1}, \ldots, \sigma_{l_{n_l}}\}$, which are elements of the vector $\boldsymbol{\sigma_l}$. The variance associated with $n_g$ inlying points is denoted as $\sigma_g^2$. Conclusively, the Huber probability density function takes the form

$$\mathbf{y}|\mathbf{f}, \sigma_g^2, \boldsymbol{\sigma}_l^2, \beta \sim \begin{cases} \prod_{i=1}^{n_g} C_1 \mathcal{N}(y_i|f(\mathbf{x}_i), \sigma_g^2) & |r_{S_i}| \leq b, \\ \prod_{i=1}^{n_l} C_2 \mathcal{N}(y_i|f(\mathbf{x}_i), \sigma_{l_i}^2) \text{Exponential}(\sigma_{l_i}^2, \beta) & |r_{S_i}| > b, \end{cases} \tag{11}$$

where $n_g + n_l = n$ is the total number of points in the training dataset. An alternative representation of the likelihood function is given by

$$\mathbf{y}_g, \mathbf{y}_l|\mathbf{f}_g, \mathbf{f}_l, \sigma_g^2, \boldsymbol{\sigma}_l^2 \sim \mathcal{N}\left(\begin{bmatrix} \mathbf{y}_g|\mathbf{f}_g \\ \mathbf{y}_l|\mathbf{f}_l \end{bmatrix}, \begin{bmatrix} \boldsymbol{\Sigma}_{gg} & \mathbf{0} \\ \mathbf{0} & \boldsymbol{\Sigma}_{ll} \end{bmatrix}\right), \tag{12}$$

where $\boldsymbol{\Sigma}_{gg}$ and $\boldsymbol{\Sigma}_{ll}$ both are diagonal matrices, the former with constant diagonal elements equal to $\sigma_g^2$ and the latter with diagonal entries $\{\sigma_{l_1}^2, \ldots, \sigma_{l_{n_l}}^2\}$. Let the hyperparameter vector $\boldsymbol{\sigma^2}$ consist of the diagonal entries of the matrix $\boldsymbol{\Sigma}_{gg}$, which are $\sigma_g^2$ and $\sigma_l^2$. The joint posterior probability density function of $\mathbf{f}$, $\sigma^2$, and $\boldsymbol{\theta}$ is given by

$$p(\mathbf{f}, \boldsymbol{\sigma}^2, \boldsymbol{\theta}) \propto p(\mathbf{y}|\mathbf{f}, \boldsymbol{\sigma}^2) p_G(\mathbf{f}|\mathbf{0}, \mathbf{K}) p(\boldsymbol{\sigma}^2|\boldsymbol{\beta}) p(\boldsymbol{\beta}|\boldsymbol{\zeta}) p(\boldsymbol{\theta}|\boldsymbol{\zeta}). \tag{13}$$

We assume that the hyper-hyperparameter vector $\boldsymbol{\beta}$ and the hyperparameter vector $\boldsymbol{\theta}$ follow the log-uniform distribution with parameters contained in $\boldsymbol{\zeta}$. Since the distribution of the variance parameter $\sigma_g^2$ of $n_g$ inlying training points is degenerate, the hyper-hyperparameter vector $\boldsymbol{\beta} = [\beta_g, \beta_l]^T$ corresponding to the $n_g$ points follows a degenerate distribution as well. Therefore, $p(\sigma_g^2|\beta_g)$ is a Dirac impulse while $\boldsymbol{\sigma}_l^2|\beta_l \sim \text{Exponential}(\boldsymbol{\sigma}_l^2|\beta_l)$. The samples generated from this distribution are highly correlated. Therefore, in order to better mix the Monte Carlo chains, we follow the trick used by Kuss (2006) as follows:

$$p(\boldsymbol{\sigma}^2, \boldsymbol{\beta}, \boldsymbol{\theta}) \propto \left[ \int p_G(\mathbf{y}|\mathbf{f}, \boldsymbol{\Sigma}) p_G(\mathbf{f}|\mathbf{0}, \mathbf{K}) d\mathbf{f} \right] p(\boldsymbol{\sigma}^2|\boldsymbol{\beta}) p(\boldsymbol{\beta}|\boldsymbol{\zeta}) p(\boldsymbol{\theta}|\boldsymbol{\zeta}), \tag{14}$$

where the covariance matrix of the $n_g$ inlying samples and the $n_l$ outlying samples is given by $\boldsymbol{\Sigma} = \begin{bmatrix} \boldsymbol{\Sigma}_{gg} & \mathbf{0} \\ \mathbf{0} & \boldsymbol{\Sigma}_{ll} \end{bmatrix}$. The samples can be used to obtain the approximated probability density functions of the latent vector function, $p(\mathbf{f}^*|\mathcal{D}, \mathbf{X}^*)$, at the new test covariates contained in $\mathbf{X}^*$ by averaging over all unknowns. Formally, we have

$$p(\mathbf{f}^*|\mathcal{D}, \mathbf{X}^*) = \int p(\mathbf{f}^*|\mathbf{f}, \boldsymbol{\sigma}^2, \boldsymbol{\theta}, \mathbf{X}^*, \mathcal{D}) p(\mathbf{f}, \boldsymbol{\sigma}^2, \boldsymbol{\theta}|\mathcal{D}) d\mathbf{f} d\boldsymbol{\sigma}^2 d\boldsymbol{\theta}. \tag{15}$$

For $T$ samples, it can be evaluated as

$$p(\mathbf{f}^*|\mathcal{D}, \mathbf{X}^*, \boldsymbol{\zeta}) = \frac{1}{T} \sum_{t=1}^{T} \int p(\mathbf{f}^*|\mathbf{f}, \mathbf{X}, \mathbf{X}^*, \boldsymbol{\theta}_t) p(\mathbf{f}|\mathcal{D}, \sigma_t^2, \boldsymbol{\theta}_t) d\mathbf{f}. \tag{16}$$

## 5.2 LAPLACE APPROXIMATION

To ensure the continuity of the derivative of the Huber density function with respect to the latent vector function $\mathbf{f}$, we utilize the pseudo-Huber loss function Charbonnier et al. (1997), which is defined as

$$\rho(r_S) = b^2 \left( \sqrt{\left(1 + \left(\frac{r_S}{b}\right)^2\right)} - 1 \right). \tag{17}$$

Laplace approximation of the posterior requires the likelihood to be log-concave in order for it to be represented by a unimodal multivariate normal distribution. It is executed by approximating the posterior distribution of $\mathbf{f}$ with a normal distribution Rue et al. (2009), that is,

$$\mathbf{f}|\mathcal{D}, \sigma, \boldsymbol{\theta} \sim \mathcal{N}(\hat{\mathbf{f}}|\mathbf{f}, \mathbf{A}). \tag{18}$$

A Taylor series expansion about the largest mode of the un-normalized posterior density function of $\mathbf{f}$ yields $q(\mathbf{f}|\mathcal{D}, \sigma, \boldsymbol{\theta}) \approx p_H(\mathbf{y}|\mathbf{f}, \sigma) p_G(\mathbf{f}|\mathbf{0}, \mathbf{K})$. The latter is used to define the MAP estimate $\hat{\mathbf{f}}$, given by

$$\hat{\mathbf{f}} = \arg \max_{\mathbf{f}} \ln q(\mathbf{f}|\mathcal{D}, \sigma, \boldsymbol{\theta}), \tag{19}$$

which may converge to a local mode in case of multimodal likelihood. As for the posterior covariance matrix, $\mathbf{A}$, it is given by

$$\mathbf{A} = (\mathbf{K}^{-1} + \mathbf{W})^{-1}, \tag{20}$$

where $\mathbf{W} = -\nabla\nabla_{\mathbf{f}} \ln \left( p_H(\mathbf{y}|\hat{\mathbf{f}}, \sigma) \right)$. The hyperparameter vector $(\sigma, \boldsymbol{\theta})$ is estimated by maximizing the log of the approximate evidence given by equation 8 using the gradient descent or the conjugate gradient method since the gradient can be analytically derived. Formally, we have

$$(\hat{\sigma}, \hat{\boldsymbol{\theta}}) = \arg \max_{(\sigma, \boldsymbol{\theta})} \ln q(\mathcal{D}|\sigma, \boldsymbol{\theta}), \tag{21}$$

where $q(\mathcal{D}|\sigma, \boldsymbol{\theta}) \approx p(\mathcal{D}|\sigma, \boldsymbol{\theta})$ is the approximate log evidence given by

$$\ln q(\mathcal{D}|\sigma, \boldsymbol{\theta}) = \ln p_H(\hat{\mathbf{f}}|\mathbf{f}) - \frac{1}{2}\ln|\mathbf{K}| - \frac{1}{2}\mathbf{f}^T\mathbf{K}^{-1}\mathbf{f} + \frac{1}{2}\ln|\mathbf{A}|. \tag{22}$$

|  | SCtMCMC | tLA | HuberMCMC | HuberLA | RCGP | GP | LaplaceMCMC |
|---|---|---|---|---|---|---|---|
| | | | $\varepsilon \sim \mathcal{N}(0.01, 0.08)$ | | | | |
| RMSE | 0.74 (0.52) | 0.75 (1.31) | **0.37** (0.42) | 0.25 (**0.25**) | 1.84 (0.82) | 1.44 (0.90) | 0.43 (0.46) |
| MAE | 0.47 (0.25) | 0.48 (0.61) | **0.31** (0.25) | 0.14 (**0.14**) | 1.28 (0.54) | 1.24 (0.68) | 0.33 (0.26) |
| | | | $\varepsilon \sim$ Student-$t(10)$ | | | | |
| RMSE | 4.86 (11.56) | 1.22 (1.31) | **0.50** (0.81) | 1.17 (**0.37**) | 1.89 (0.88) | 1.52 (0.98) | 0.59 (0.93) |
| MAE | 1.67 (1.25) | 0.77 (0.65) | **0.41** (0.39) | 0.79 (**0.18**) | 1.71 (0.85) | 1.34 (0.22) | 0.43 (0.35) |
| | | | $\varepsilon \sim$ Laplace$(0, 0.1)$ | | | | |
| RMSE | 4.76 (0.48) | 1.23 (1.31) | **0.58** (0.42) | 1.17 (**0.35**) | 1.95 (0.86) | 1.51 (0.89) | 1.06 (0.82) |
| MAE | 1.64 (0.23) | 0.76 (0.61) | **0.41** (0.24) | 0.68 (**0.18**) | 1.27 (0.46) | 1.23 (0.41) | 0.75 (0.34) |
| | | | $\varepsilon \sim$ Student-$t(1)$ (Cauchy) | | | | |
| RMSE | 4.75 (0.57) | 1.25 (1.32) | **0.61** (0.49) | 1.20 (**0.17**) | 1.97 (0.62) | 1.50 (0.89) | 0.42 (0.75) |
| MAE | 1.65 (0.27) | 0.78 (0.67) | **0.47** (0.27) | 0.81 (**0.11**) | 1.78 (0.42) | 1.32 (0.66) | 0.66 (0.38) |

Table 1: RMSE and MAE values on the Neal dataset for the Case 1. Values in parentheses represent the performance for Case 3. Bold values highlight the best performance with the lowest RMSE and MAE.

## 6 EXPERIMENTS

Through our experiments, we aim to address the following questions:

($Q$1) When is HuberLA (GP-Huber with Laplace's method) preferable, and under which outlier scenarios is HuberMCMC (GP-Huber with Gibbs sampling) more suitable?

($Q$2) Does GP-Huber show a significant performance improvement over standard GP regression and the RCGP method proposed by Altamirano et al. (2024)?

($Q$3) Does the use of projection pursuit weighting offer a tangible advantage?

($Q$4) Does GP-Huber provide more accurate estimates of the planet-to-star radius ratio compared to the standard GP method used by Gibson et al. (2012) in the transmission spectroscopy experiment?

We performed extensive experiments on benchmark datasets, considering cases of extreme outliers based on their locations, magnitudes, and various error distributions. The threshold $b$ was set to $1.5$ for Gaussian error distributions and $0.45$ for Student's-t, Laplace, and Cauchy distributions. For all experiments, including the transmission spectroscopy, we used a anisotropic squared exponential kernel function. The mean function is assumed to be zero except for the spectroscopy experiment. Performance was evaluated using root mean square error (RMSE) and mean absolute error (MAE) metrics.

### 6.1 NEAL DATASET

We evaluate the proposed GP-Huber on the Neal dataset (Neal, 1997) for the following cases of extreme outliers:

**Case 1:** Extreme outliers $y_i^{(l)}$, $\boldsymbol{x}_j^{(l)}$ in added in output and covariate dimensions, respectively.

**Case 2:** Only output dimensions $y_i^{(l)}$ were contaminated with extreme data points.

**Case 3:** Bad data points $y_i^{(c)}$, $\boldsymbol{x}_j^{(l)}$ in added to both output and covariate dimensions, respectively, with the former being relatively close to the main data cluster compared to Case 1.

**Case 4:** Only output dimensions were contaminated with data points $y_i^{(c)}$ relatively close to the data cloud compared to Case 1.

In all the cases above, the locations $i$ and $j$ of the output and covariate outliers may differ or coincide (refer to Appendix B.1 for the location and magnitude details on outliers). For each case, we considered four different error distributions: $\mathcal{N}(0.01, 0.08)$, Student-t$(10)$, Laplace$(0, 0.1)$, Student's-t$(1)$.

The baseline models considered for comparison on the Neal dataset, along with RCGP, include: GP with a Student's t error model solved using MCMC integration (SCtMCMC), GP with a Student's t error model using Laplace approximation (tLA), and GP with a Laplace likelihood solved via MCMC integration (LaplaceMCMC). Table 1 presents the RMSE and MAE values comparing GP-Huber against these baselines for Cases 1 and 3. Refer to Appendix B.1 for the Tables 5, 6 for the Cases 2, 4. Now, we are in position to answer $Q$1.

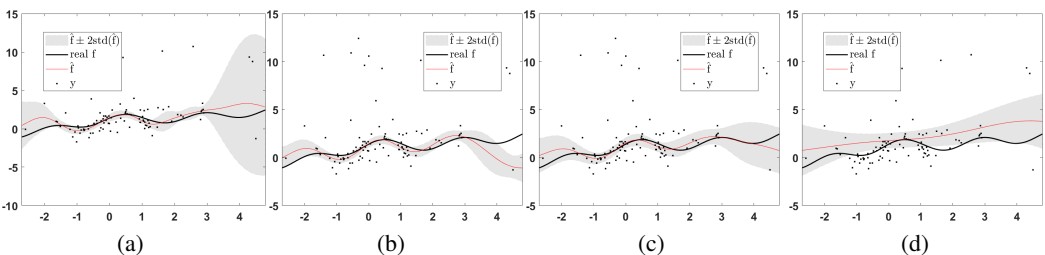

Figure 2: Predicted values for the Case 1 of the Student's t-error distribution for the Neal dataset obtained from the eight considered GP regression models: (a) SCtMCMC; (b) tLA ; (c) HuberLA; (d) GP.

**When is HuberMCMC better?**

In scenarios with $y^{(l)}, \boldsymbol{x}^{(l)}$ (Case 1), HuberMCMC performed better than HuberLA (see, Tables 1 and 5). HuberMCMC also outperformed tLA in predictive accuracy, demonstrating a more robust fit that is less influenced by $\boldsymbol{x}^{(l)}$ (Figure 2). HuberLA generally provided better uncertainty quantification compared to HuberMCMC (see Figures 2 and 5), while maintaining competitive predictive performance. In outlier scenarios with $y^{(l)}$ (Case 2), HuberMCMC exhibited superior performance across Student's-t, Laplace, and Cauchy error distributions (see, Table 5). This suggests that HuberMCMC is a robust choice for datasets containing extreme output outliers i.e. outlier scenarios similar to Cases 1 and 2.

**When is HuberLA better?**

HuberLA exhibited superior performance in handling closer output outliers $y^{(c)}$ compared to HuberMCMC (values in parenthesis in the Table 1 and Table 6). Figure 6 highlights HuberLA's robustness to , in contrast to tLA which is influenced by such points. While HuberLA generally provided more accurate predictions and reliable uncertainty quantification than both HuberMCMC and tLA, HuberMCMC performed competitively for the Cases 3 and 4.

When we did not add $\boldsymbol{x}^{(l)}$ (Cases 2 and 4), HuberLA and HuberMCMC exhibited performance comparable to other baselines, indicating their robustness to exclusively $y^{(l)}$ and $y^{(c)}$. In this case, we did not need to apply projection pursuit weighting on $\boldsymbol{x}$. However, the RMSE and MAE values in Table 1 (with projection pursuit weighting) are clearly lower than those in Tables 5 and 6 (without weighting). This demonstrates that the weighting mechanism enhances GP-Huber's accuracy, addressing $\mathcal{Q}3$. Compared to the RCGP, both HuberLA and HuberMCMC consistently produced better predictive performance across all outlier cases and error distributions applied to the Neal dataset. (Please refer to Appendix B.1 for Figures 5, 6.)

## 6.2 UCI DATASETS

In this set of experiments, we compared the performance of GP-Huber on the UCI datasets, Energy and Yacht, against RCGP and other baselines: t-GP, m-GP, and standard GP, using the outlier settings from Altamirano et al. (2024). We specifically focused on the "focused outlier" and "asymmetrical outlier" scenarios, as they closely resemble our extreme and close outlier cases.

|  | GP | RCGP | t-GP | m-GP | HuberMCMC | HuberLA |
|---|---|---|---|---|---|---|
| | | | Focused Outliers | | | |
| Energy | 0.03 | **0.02** | 0.03 | 0.24 | 0.12 | 0.04 |
| Yacht | 0.26 | **0.10** | 0.20 | 0.24 | 0.37 | 0.28 |
| | | | Asymmetric Outliers | | | |
| Energy | 0.54 | 0.44 | 0.42 | 0.41 | **0.06** | 0.07 |
| Yacht | 0.54 | 0.35 | 0.41 | 0.40 | **0.29** | 0.42 |

Table 2: MAE values for energy and yacht. Bold values indicate the best performance for each row.

MAE values of the comparison are presented in Table 2. As expected, HuberLA demonstrates to be more robust than HuberLA since the asymmetrical and focused outliers cases considered in the study of Altamirano et al. (2024) broadly fall under the Cases 3 and 4 in our study. On the Energy dataset, HuberLA outperformed both tLA and RCGP. On the twitter flash crash dataset, HuberLA outperforms RCGP in both RMSE and MAE (see Table 3).

| | GP | RCGP | HuberMCMC | HuberLA | | RCGP | HuberMCMC | HuberLA |
|---|---|---|---|---|---|---|---|---|
| RMSE | 0.354 | 0.331 | 0.0118 | **0.0021** | Flashcrash | 8.71 | 26.6 | **3.41** |
| MAE | 0.154 | 0.124 | 0.0089 | **0.0014** | Neal | 4.47 | 6.28 | **2.73** |

Table 3: RMSE and MAE for Twitter flash crash.  Table 4: Processing times (in seconds).

HuberMCMC (Gibbs sampling) and HuberLA (Laplace approximation) have similar computational times to RCGP. HuberLA consistently converged within 2 to 4 seconds, while HuberMCMC showed more variability, with times ranging from 5 to 30 seconds. Table 4 shows the processing times for HuberLA and HuberMCMC compared to RCGP on the Twitter flash crash and Neal datasets for Case 1. In our experiments, both HuberMCMC and HuberLA outperformed RCGP and standard GP, with HuberLA showing the best computational efficiency, thus answering $\mathcal{Q}2$.

## 6.3 Transmission Spectroscopy

Transmission spectroscopy records the relative change in the stellar flux, which is the incident photons per unit area, as a planet travels in front of the star. The sources of error, such as photon noise and instrumental and astrophysical systematics, raise many potential challenges for precise planet's atmosphere characterization. The goal is to infer the planet to star radius ratio $\rho_{radius}$ from the observed flux as the planet passes in front of the star. The optical state parameters are metered via auxiliary measurements of the spectral trace such as position, width, angle, or other parameters, indicating the state of the detector and optics, which are thought to be the cause of instrumental systematics. Instead of modeling the latter as a linear function of the optical state parameters, Gibson et al. (2012) proposed a non-parametric model by leveraging GPs.

The observation set obtained from HST-NICMOS includes the light curves for 18 wavelength channels extracted from $n = 638$ spectra of the planetary system HD-189733. The flux measurements contained in the vector, $\boldsymbol{f} = [f_1, f_2, \ldots, f_n]^T$, are recorded at $n$ time instants, $\{t_1, t_2, \ldots, t_n\}$ and the optical state parameters $\mathbf{x}_{t_i}$ collected in the matrix $\mathbf{X} \in \mathbb{R}^{n \times d}$ constitute the training dataset. We extend the work of Gibson et al. (2012) by using the GP-Huber model to estimate the planet-to-star radius ratio $\rho_{radius}$. As demonstrated earlier, the robustness to outliers of GP-Huber allows us to utilize 517 measurements associated with four out-of-transit orbits, namely orbit numbers, $\{2, 3, 4, 5\}$, and 137 measurements associated with one in-transit orbit, namely orbit number 1. The latter was excluded from the analysis performed by Gibson et al. (2012) as it constitutes much larger systematics effects attributed to the spacecraft settling. The observed transit flux modeled in the GP framework follows a normal distribution, that is,

$$\boldsymbol{f}(\boldsymbol{t}, \mathbf{X}) \sim \mathcal{N}(\boldsymbol{T}(\boldsymbol{t}, \boldsymbol{\phi}), \mathbf{K}), \tag{23}$$

where the parameter vector, $\boldsymbol{\phi}$, include the parameter of interest, $\rho_{radius}$, and other parameters. We consider the analytical quadratic limb darkening transit function proposed by Mandel & Agol (2002). Analogous with equation 11, we assume that the observed transit flux vector, $\boldsymbol{f} = \boldsymbol{f}(\boldsymbol{t}, \mathbf{X})$, in the GP-Huber framework follows a normal distribution, that is,

$$\boldsymbol{f} | \boldsymbol{T}(\boldsymbol{t}, \boldsymbol{\phi}), \mathbf{X}, \boldsymbol{\phi}, \boldsymbol{\theta}, \boldsymbol{\sigma}^2 \sim \mathcal{N}\left(\boldsymbol{T}(\boldsymbol{t}, \mathbf{X}), \boldsymbol{\Sigma} + \mathbf{K}\right). \tag{24}$$

The joint un-normalized log-posterior function of $\boldsymbol{\phi}$, $\boldsymbol{\beta}$, and $\boldsymbol{\theta}$ with the gamma aprior probability density function, $p(\boldsymbol{\theta}) = \frac{1}{l}\exp\left(\frac{-\boldsymbol{\theta}}{l}\right)$, over the covariance function hyperparameters is given by

$$\log P(\boldsymbol{\phi}, \boldsymbol{\theta}, \boldsymbol{\sigma}^2, \boldsymbol{\beta} | \boldsymbol{f}, \mathbf{X}, \boldsymbol{\zeta}) = \log\left(\mathcal{L}(\mathbf{r}_S | \mathbf{X}, \boldsymbol{\phi}, \boldsymbol{\theta}, \boldsymbol{\sigma}^2)\right) - \frac{\tau}{l_\tau} - \sum_{i=1}^d \left(\frac{1}{s_i l_i}\right) + \log(\boldsymbol{\beta}) - \boldsymbol{\beta}^T \boldsymbol{\sigma}^2 + \log(p(\boldsymbol{\beta} | \boldsymbol{\zeta})) + \mathrm{C}. \tag{25}$$

The challenging task now is to infer the parameter $\rho_{radius}$ from the joint posterior distribution of $(\boldsymbol{\phi}, \boldsymbol{\theta}, \boldsymbol{\sigma}^2, \boldsymbol{\beta})$. The log-likelihood $\mathcal{L}$ term is expressed as

$$\log \mathcal{L}(\boldsymbol{r}_S | \mathbf{X}, \boldsymbol{\phi}, \boldsymbol{\theta}, \boldsymbol{\sigma}^2) = \frac{-1}{2}\mathbf{r}_S^T(\boldsymbol{\Sigma} + \mathbf{K})^{-1}\mathbf{r} - \frac{1}{2}\log|\boldsymbol{\Sigma} + \mathbf{K}| - \frac{n}{2}\log(2\pi) + \log(1 - \varepsilon), \tag{26}$$

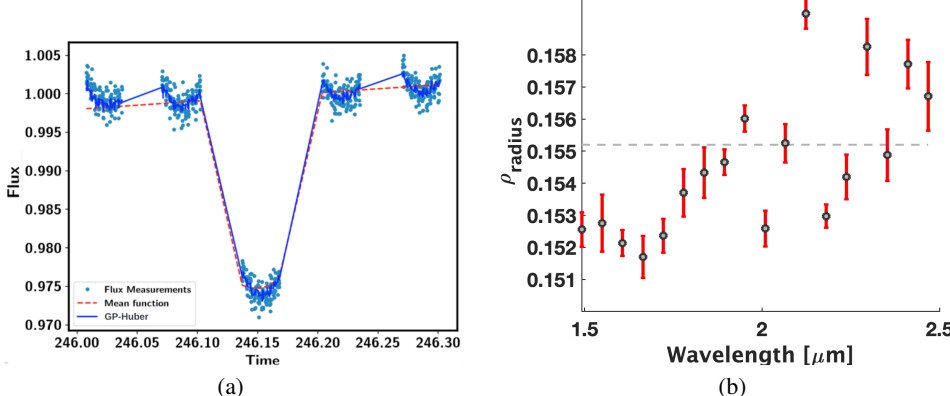

(a)

(b)

Figure 3: Transit curve fit and estimated $\rho_{radius}$. $(a)$ Transit curve mean function $T(t, \boldsymbol{\theta})$ and GP-Huber model fit; $(b)$ results of planet-to-star radius ratios ($\rho_{radius}$) obtained from GP-Huber with error-bars.

where $\mathbf{r} = \boldsymbol{f} - \boldsymbol{T}(\boldsymbol{t}, \mathbf{X})$. One of the approaches is to use the Bayesian method that seeks the posterior distribution of $\rho_{radius}$ by marginalizing over the other parameters of the mean function parameters $\phi$ and the covariance function hyperparameters, $\boldsymbol{\theta}$ using MCMC methods. The other method proposed as the type-II maximum likelihood method by Gibson et al. (2012), where the hyperparameters, $\boldsymbol{\theta}$ and $\boldsymbol{\sigma}^2$. Formally, we have

$$(\hat{\boldsymbol{\phi}}, \hat{\boldsymbol{\theta}}, \hat{\boldsymbol{\sigma}}^2, \hat{\boldsymbol{\beta}}) = \arg\max_{\boldsymbol{\phi}, \boldsymbol{\theta}, \boldsymbol{\sigma}^2, \boldsymbol{\beta}} \log P(\boldsymbol{\phi}, \boldsymbol{\theta}, \boldsymbol{\sigma}^2, \boldsymbol{\beta} | \boldsymbol{f}, \mathbf{X}, \boldsymbol{\zeta}). \tag{27}$$

And the posterior distribution of the parameter of interest $\rho_{radius}$ is obtained by marginalizing the joint posterior distribution $p(\boldsymbol{\phi}, \boldsymbol{\theta}, \boldsymbol{\sigma}^2, \boldsymbol{\beta})$ over the hyperparameters and the rest of the mean function parameters. In the standard type II maximum likelihood method, the hyperparameters are fixed to their maximum likelihood estimates i.e. by maximizing the evidence $p(\mathcal{D}|\boldsymbol{\phi}, \boldsymbol{\theta}, \boldsymbol{\sigma}^2)$.

Figure 3(a) shows the transit fit obtained for one wavelength channel. Figure 3(b) shows the estimated $\rho_{radius}$ obtained using MCMC integration over the rest of the mean function parameters $\phi$ and hyperparameters $\boldsymbol{\theta}$ along with the values estimated from the white light curve represented as the white dashed line. Note that the estimated $\rho_{radius}$ values are very close to the white light curve value of $0.155$. Most of our results agree with the results obtained from the Gibson model except for wavelength channels $1.665\mu$m and $2.124\mu$m (see, Appendix B.2), which effectively answers $Q4$.

Our code[1] was implemented in Matlab R2023a with the help of package gpstuff on Intel i7.

## 7 CONCLUSIONS

The proposed GP-Huber model shows promise for handling a variety of heavy-tailed and Gaussian error distributions with extreme outliers in both covariate and output dimensions. Notably, it introduces no extra parameters to infer. The model's unimodal posterior simplifies Gibbs sampling and allows for an efficient Laplace approximation. From our experiments on the Neal and UCI datasets, we observe that HuberMCMC and HuberLA offer superior robustness compared to RCGP and other baselines. Additionally, the transmission spectroscopy experiment demonstrates their potential in real-world applications.

In future work, we will examine GP-Huber's performance with skewed error distributions and investigate the use of high breakdown estimators for highly corrupted real-world datasets. Another direction for future work involves extending the scalability of GP-Huber to handle large datasets by implementing sparse inference techniques.

---

[1]https://anonymous.4open.science/r/GpHuber-6A2D

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

## A    PROOFS AND THEORETICAL EXPLANATIONS

### A.1    PROOF OF THEOREM 1

*Proof.* The derivative of the log-posterior on $f$ is proportional to

$$h_i(f_i) = \frac{-(y_i - f_i)}{\sqrt{1 + (y_i - f_i)^2}} - v_i, \tag{28}$$

where $v_i$ is the $i^{\text{th}}$ term of $\mathbf{v} = \mathbf{K}^{-1}\mathbf{f}$. The posterior distribution is unimodal if there exist a real solution to $h(f_i)$. The first part of equation 28 is strictly increasing in each $f_i$ upon the first derivative test. Its behaviour at limits is

$$\frac{-(y_i - f_i)}{\sqrt{1 + (y_i - f_i)^2}} = \begin{cases} 0 & \text{when } f_i \to y_i, \\ -1 & \text{when } f_i \to \infty. \end{cases}$$

As for the second part $\mathbf{v}$, the symmetric matrix $\mathbf{K}^{-1}$ can be diagonalized and its action can be understood in terms of its eigenvalues and eigenvectors. The term is linear and thus continuously maps $\mathbf{f}$ to $\mathbb{R}^n$. By the intermediate value theorem and the strict monotonicity, this equation has a unique and real solution. $\qquad\square$

## A.2 SELECTION OF THE THRESHOLD B

The Huber estimator is a maximum likelihood estimator associated with the least favorable density function given by

$$\tilde{g}(r) = \frac{1-\varepsilon}{\sqrt{2\pi}\sigma}e^{-\rho\left(\frac{r}{\sigma}\right)}, \tag{29}$$

which can be further elaborated as

$$\tilde{g}(r) = \begin{cases} \frac{1-\varepsilon}{\sqrt{2\pi}}e^{-\frac{r^2}{2}} & \text{for } |r| \le b \\ \frac{1-\varepsilon}{\sqrt{2\pi}}e^{-|b||r|-\frac{b^2}{2}} & \text{for } |r| > b \end{cases} \tag{30}$$

This distribution is Gaussian in the center and Laplacian in the tails. The threshold $b$ is related to the fraction of contamination $\varepsilon$ against which we want to be protected. This relation is obtained by setting

$$\int_{-\infty}^{\infty} \tilde{g}(r)\,dr = 1 \tag{31}$$

yielding

$$\int_{-b}^{b} \frac{(1-\varepsilon)}{\sqrt{2\pi}}e^{-\frac{r^2}{2}}\,dr + 2(1-\varepsilon)\int_{b}^{\infty}\frac{1}{\sqrt{2\pi}}e^{\left(-br+\frac{b^2}{2}\right)}\,dr = 1 \tag{32}$$

$$(1-\varepsilon)\int_{-b}^{b}\frac{1}{\sqrt{2\pi}}e^{-\frac{r^2}{2}}\,dr = (1-\varepsilon)[1-2(1-\Phi(b))] = (1-\varepsilon)(2\Phi(b)-1); \tag{33}$$

and

$$2(1-\varepsilon)\int_{b}^{\infty}\frac{1}{\sqrt{2\pi}}e^{\left(-br+\frac{b^2}{2}\right)}\,dr = 2(1-\varepsilon)\frac{1}{\sqrt{2\pi}}\left[-\frac{1}{b}e^{\left(-br+\frac{b^2}{2}\right)}\right]_{b}^{\infty} \tag{34}$$

$$= \frac{2(1-\varepsilon)}{\sqrt{2\pi}}\frac{1}{b}e^{-\frac{b^2}{2}} = \frac{2(1-\varepsilon)}{b}\phi(b); \tag{35}$$

Solving further, we get

$$2\Phi(b) - 1 + \frac{2}{b}\Phi(b) = \frac{1}{1-\varepsilon} \tag{36}$$

We observe that $b$ decreases to 0 as $\varepsilon$ increases to 1. At $b = 1.5$, the Huber loss can handle roughly $\varepsilon = 0.1$ i.e. 10% of outliers.

# B  ADDITIONAL EXPERIMENTS

## B.1  NEAL DATASET

(Neal, 1997) proposed the following artificial model:

$$g(\boldsymbol{x}_i) = 0.3 + 0.4x + 0.5sin(2.7x) + 1.1/(1+x^2). \tag{37}$$

A sample of $n = 100$ points constitutes the training data set, $(\mathbf{X}, \mathbf{y})$. The predictions of the vector function, $\mathbf{f}^*$, are made at $n^* = 541$ test covariates contained in $\boldsymbol{x}^*$ over the interval $[-2.7, 5]$. Since the projection statistics require at least a two-dimensional covariate space, they are calculated on the regressors' vector, $\mathbf{x}$ combined with the column of ones, i.e., on the matrix $\mathbf{H} = [\mathbf{1}, \mathbf{x}]$. Specifically, for a test point $\mathbf{h}_i = [1, x_i]$, PS$(\mathbf{h}_i)$ is calculated using equation 4 in the paper. The training covariate, $x_i$, is flagged as an outlier if the associated projection pursuit weights, $w_i = \min\left(1, \frac{c}{\text{PS}(\mathbf{h}_i)^2}\right)$, has a value less than one.

We demonstrate the proposed GP-Huber in four cases of error probability distribution: (i) $\mathcal{N}(0.01, 0.08)$; (ii) the Student's t-distribution with 10 degrees of freedom; (iii) Laplace$(0, 0.1)$; and (iv) the Cauchy distribution. For each of these error distributions, we introduce extreme output outliers $y^l = \{90.5, 8.6, 98.1, 5.3, 5.2, 6.1, 1, 8\}$ at locations $j = \{7, 8, 9, 10, 11, 15, 61, 70\}$, extreme covariate data points $x^{(l)} = \{4.3, 4.4, 4.5\}$ at locations $i = \{21, 22, 23\}$. We also add large

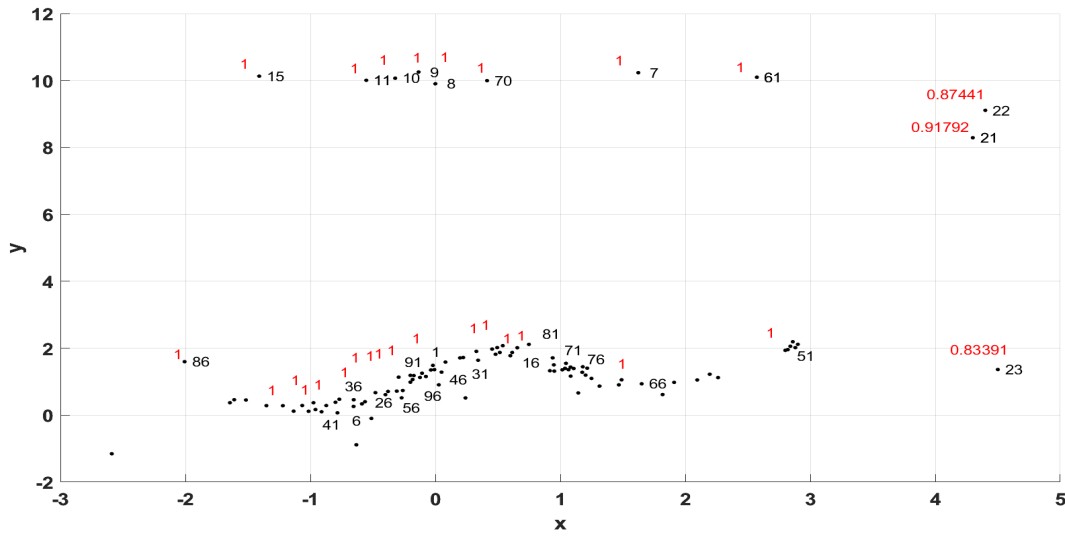

Figure 4: Weights based on PS for the Neal data. The numbers right to the data points indicate index numbers and the ones to the left in red color indicate the weights associated with that data point.

magnitudes to introduce group of good data points to the covariates $\{x_{50}, x_{51}, x_{52}, x_{53}, x_{54}, x_{55}\}$ for which $y_i = g(\boldsymbol{x}_i)$.

We observe that the projection pursuit weights based on the PS corresponding to the bad leverage points are $\{0.9179, 0.8744, 0.8339\}$ while those corresponding to the good leverage points are equal to 1 (see Figure 4).

|  | SCtMCMC | tLA | HuberMCMC | HuberLA | RCGP | GP | LaplaceMCMC |
|---|---|---|---|---|---|---|---|
| $\varepsilon \sim \mathcal{N}(0.01, 0.08)$ | | | | | | | |
| RMSE | 1.41 | 1.30 | 1.40 | 1.36 | 2.04 | 1.74 | 1.48 |
| MAE | 0.90 | 0.81 | 0.99 | 0.95 | 1.93 | 1.51 | 0.98 |
| $\varepsilon \sim \text{Student-t}(10)$ | | | | | | | |
| RMSE | 1.22 | 1.14 | 0.91 | 1.12 | 2.04 | 1.66 | 1.01 |
| MAE | 0.63 | 0.56 | 0.62 | 0.67 | 1.85 | 1.34 | 0.92 |
| $\varepsilon \sim \text{Laplace}(0, 0.1)$ | | | | | | | |
| RMSE | 1.38 | 2.73 | 1.33 | 1.37 | 2.06 | 1.73 | 1.33 |
| MAE | 0.88 | 1.82 | 0.97 | 0.96 | 1.95 | 1.51 | 0.95 |
| $\varepsilon \sim \text{Student-t}(1)$ (Cauchy) | | | | | | | |
| RMSE | 4.74 | 2.11 | 1.33 | 1.38 | 2.11 | 1.75 | 1.33 |
| MAE | 1.67 | 1.36 | 0.96 | 0.98 | 1.84 | 1.50 | 0.95 |

Table 5: Results for Case 2

## B.2 TRANSMISSION SPECTROSCOPY

Transmission spectroscopy records the relative change in the stellar flux, which is the incident photons per unit area, as a planet travels in front of the star around which it revolves. When the planet faces the star directly, known as a transit, it occludes a fraction of the stellar flux emitted by the star equal to the sky-projected area of the planet as compared to the area of the star, which is referred to as transit depth. The measurement of the total flux over time is known as the light curve. The property on which the transmission spectroscopy relies to estimate the transit curve parameters is the planet's transit depth, which dependents on the wavelengths of the transmitted flux. For the wavelengths where the planet's atmosphere is opaque due to the absorption of the emitted electromagnetic waves by constituent atoms or molecules, the planet blocks slightly more stellar flux. The variations are

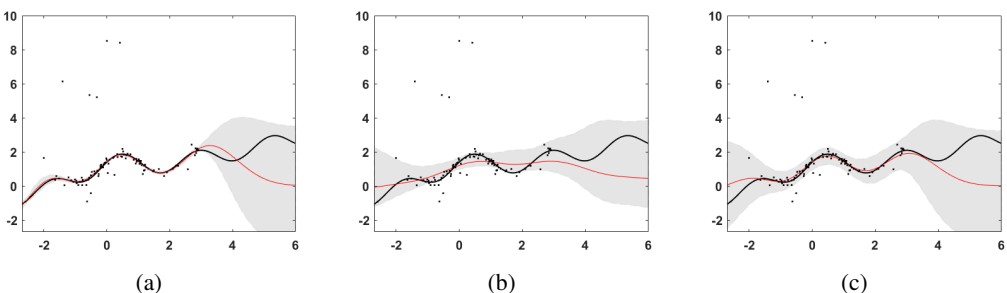

Figure 5: Predicted values for (a) tLA; (b) HuberMCMC; (c) HuberLA with standard deviations for the Case 2 with error following Student's t distribution on Neal dataset.

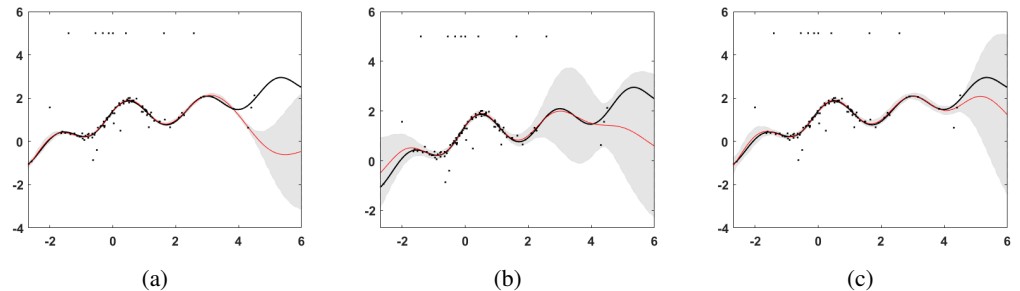

Figure 6: Predicted values for (a) tLA; (b) HuberMCMC; (c) HuberLA with standard deviations for the Case 3 with error following Student's t distribution on Neal dataset.

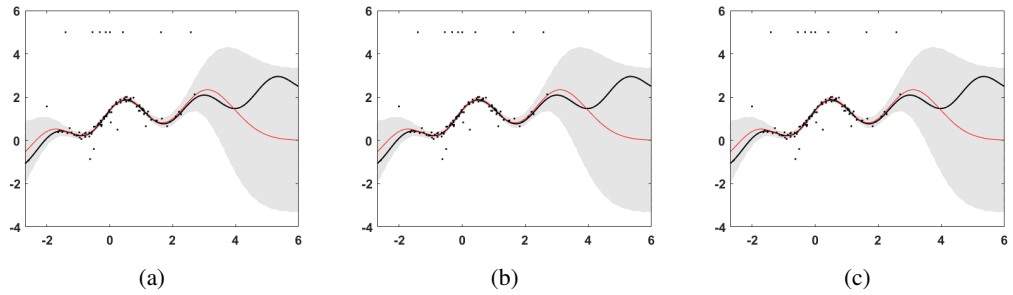

Figure 7: Predicted values for (a) tLA; (b) HuberMCMC; (c) HuberLA with standard deviations for the Case 4 with error following Student's t distribution on Neal dataset.

| | SCtMCMC | tLA | HuberMCMC | HuberLA | RCGP | GP | LaplaceMCMC |
|---|---|---|---|---|---|---|---|
| | | | $\varepsilon \sim \mathcal{N}(0.01, 0.08)$ | | | | |
| RMSE | 1.02 | 1.01 | 1.48 | 1.50 | 1.10 | 1.17 | 0.98 |
| MAE | 0.51 | 0.52 | 0.79 | 0.54 | 0.76 | 0.78 | 0.53 |
| | | | $\varepsilon \sim \text{Student-t}(10)$ | | | | |
| RMSE | 1.58 | 1.02 | 1.17 | 1.13 | 1.11 | 1.17 | 0.61 |
| MAE | 1.28 | 0.52 | 0.53 | 0.78 | 0.76 | 0.85 | 0.35 |
| | | | $\varepsilon \sim \text{Laplace}(0, 0.1)$ | | | | |
| RMSE | 1.04 | 1.01 | 1.06 | 1.18 | 1.16 | 1.08 | 1.16 |
| MAE | 0.51 | 0.52 | 0.53 | 0.78 | 0.66 | 0.66 | 0.58 |
| | | | $\varepsilon \sim \text{Student-t}(1)$ (Cauchy) | | | | |
| RMSE | 1.58 | 1.02 | 1.18 | 1.02 | 1.10 | 1.07 | 1.04 |
| MAE | 1.28 | 0.52 | 0.63 | 0.78 | 0.56 | 0.56 | 0.52 |

Table 6: Neal results for the Case 4.

measured by binning the light curve into spectrophotometric channels of different wavelengths and by fitting the light curve from each channel separately with a transit model Kreidberg (2017).

The sources of error, such as photon noise and instrumental and astrophysical systematics, raise many potential challenges for precise atmosphere characterization. Pointing drift or modifications in the telescope focus influence the spectrum position on the detector to a small degree during transit due to instrumental systematics. Note that instrumental systematics are nothing but what is popularly known as systematic errors in statistics, which are here attributed to the atmospheric effects on the physical properties of an instrument. The optical state parameters are metered via auxiliary measurements of the spectral trace such as position, width, angle, or other parameters, indicating the state of the detector and optics, which are thought to be the cause of instrumental systematics. Instead of modeling the latter as a linear function of the optical state parameters, Gibson et al. (2012) proposed a non-parametric model by leveraging GPs.

The observation set obtained from HST- NICMOS includes the light curves for 18 wavelength channels extracted from n=638 spectra along with six optical state parameters, namely the position of the spectral trace along the dispersion axis ,$\Delta X$, the average position of the spectral trace along the cross-dispersion axis, $\Delta Y$, the angle of the spectral trace with the x-axis, $W$, the average width of the spectral trace, $\psi^s$, the temperature, $T$, and the orbital phase, $\psi^H$. The flux measurements contained in the vector, $\boldsymbol{f} = [f_1, f_2, \ldots, f_n]^T$, are recorded at $n$ time instants, $\{t_1, t_2, \ldots, t_n\}$, contained in the time vector, $\boldsymbol{t}$, and the optical state parameters are given by $\mathbf{x}_i = [\Delta X_i, \Delta Y_i, W_i, \psi_i^H, T_i, \psi_i^s]^T$ at time instant, $t_i$, collected in the matrix $\mathbf{X} \in \mathbb{R}^{6 \times N}$ given by $\mathbf{X} = [\mathbf{x}_1, \ldots, \mathbf{x}_n]$.

The observed transit flux modeled in the GP framework follows a normal distribution, that is,

$$\boldsymbol{f}(\boldsymbol{t}, \mathbf{X}) \sim \mathcal{N}(\boldsymbol{T}(\boldsymbol{t}, \boldsymbol{\phi}), \mathbf{K}(\mathbf{X}, \mathbf{X}|\boldsymbol{\theta})). \tag{38}$$

where the parameter vector, $\boldsymbol{\phi}$, include the parameter of interest, $\rho_{radius}$, and other parameters, namely out-of-transit flux, $f_{oot}$, time gradient, $T_{grad}$, fixed central transit time, $T_0$, orbital period, $P$, limb darkening coefficient, $c_1$, limb darkening coefficient, $c_2$. The transit vector function, $\boldsymbol{T}(\boldsymbol{t}, \boldsymbol{\phi})$, is hereafter referred to as mean function parameter vector. The non-variable mean function parameters are fixed or calculated as stated in Gibson et al. (2012). Along with the planet-to-star radius ratio, the other mean function parameters are the parameters of a linear baseline model, $f_{oot}$ and $T_{grad}$. The covariance matrix, $\boldsymbol{\Sigma}(\mathbf{x}_i, \mathbf{x}_j|\boldsymbol{\theta})$, is the covariance between two output flux measurements defined as a function of the distance between optical state parameters, $(\mathbf{x}_i, \mathbf{x}_j)$, given by

$$K_{ij} = k(\mathbf{x}_i, \mathbf{x}_j) + \delta_{ij}\sigma^2, \tag{39}$$

where $k(\cdot, \cdot)$ is a Gaussian kernel. The threshold parameter, $b$, is set to 1.5 to achieve good robustness and efficiency at data distributed normally.

Table 7: Results of the planet-to-star radius ratio obtained from Gibson (2012) and GP-Huber.

| Wavelength | Results from model in Gibson2012 | | Results obtained from GP-Huber | |
|---|---|---|---|---|
| ($\mu$m) | $\rho_{radius}$ | $\Delta\rho_{radius}$ | $\rho_{radius}$ | $\Delta\rho_{radius}$ |
| 2.468 | 0.15545 | 0.00077 | 0.15525 | 0.00071 |
| 2.411 | 0.15520 | 0.00052 | 0.15771 | 0.0008911 |
| 2.353 | 0.15455 | 0.00044 | 0.15488 | 0.0004021 |
| 2.296 | 0.15513 | 0.00057 | 0.15825 | 0.0006526 |
| 2.238 | 0.15512 | 0.00041 | 0.1542 | 0.0005276 |
| 2.181 | 0.15504 | 0.00051 | 0.15297 | 0.0007462 |
| 2.124 | 0.15417 | 0.00066 | 0.15928 | 0.0007869 |
| 2.066 | 0.15508 | 0.00066 | 0.15525 | 0.000399 |
| 2.009 | 0.15393 | 0.00036 | 0.15259 | 0.0004077 |
| 1.951 | 0.15595 | 0.00051 | 0.15602 | 0.0005586 |
| 1.894 | 0.15549 | 0.0006 | 0.15466 | 0.0005988 |
| 1.837 | 0.15513 | 0.00053 | 0.15433 | 0.0004704 |
| 1.779 | 0.15534 | 0.00051 | 0.1537 | 0.0003601 |
| 1.722 | 0.15447 | 0.00087 | 0.14937 | 0.0006938 |
| 1.665 | 0.15429 | 0.00064 | 0.1517 | 0.000871 |
| 1.607 | 0.15266 | 0.00062 | 0.15213 | 0.0008045 |
| 1.55 | 0.15359 | 0.00073 | 0.15276 | 0.0007583 |
| 1.492 | 0.15367 | 0.00118 | 0.15256 | 0.0010653 |

The joint un-normalized log-posterior function of $\phi$, $\beta$, and $\theta$ is given by

$$\log P(\phi, \theta, \sigma^2, \beta | f, \mathbf{X}, \zeta) = \log \left( \mathcal{L}(\mathbf{r}|\mathbf{X}, \phi, \theta, \sigma^2) \right) - \frac{\tau}{l_\tau} - \sum_{i=1}^{d} \left( \frac{1}{s_i l_i} \right)$$
$$+ \log(\beta) - \beta^T \sigma^2 + \log(p(\beta|\zeta)) + \text{C}. \quad (40)$$

Here, we lay the gamma a priori probability density function, $p(\theta) = \frac{1}{l}\exp\left(\frac{-\theta}{l}\right)$ over the covariance function hyperparameters $\theta$. The parameter $l_\tau$ is of the gamma a priori associated with hyperparameter $\tau$ and C represents additional constant terms. The samples of $\beta_l$ are generated from log uniform distribution to lay a non-informative prior with parameter vector, $\zeta$, whereas $p(\beta_g)$ is a degenerate probability density function.

The values of the planet-to-star radius ratio $\rho_{radius}$ for each wavelength obtained from the GP-Huber model are shown in Table 7 along with those obtained from the model described in Gibson et al. (2012) referred to as Gibson2012, where $\Delta\rho_{radius}$ represents the estimated uncertainty.

