# OpenReview forum: "Robust Gaussian Process Regression with Huber Likelihood"
_ICLR.cc/2025/Conference — ICLR 2025 Conference Withdrawn Submission_

### Official Review · Reviewer_rnXe · 2024-10-16

**Soundness:** 2
**Presentation:** 1
**Contribution:** 2
**Rating:** 3
**Confidence:** 5

**Summary:**

This paper considers the problem of performing GP regression in the context of likelihood misspecification. More specifically, the paper considers the use of two tricks: (i) replacing the Gaussian likelihood with a Huber likelihood, (ii) standardising residuals to reduce the impact of outliers in covariate space. Combined, these trick lead to a method which seem to perform well (in terms of accuracy) on problems where outliers are present in covariate space. Unfortunately, the use of the Huber likelihood breaks conjugacy, and so the authors propose two approximations of their posterior: one based on MCMC and the other based on a Laplace approximation.

**Strengths:**

I found it particularly interesting that the present method can deal with outliers in covariate space as well as outliers in the response. As the authors correctly point out, this is something that some of the existing methods really struggle with.

**Weaknesses:**

Presentation:
- Overall, I found the paper to be poorly written. There are numerous times where there are clearly missing words or sentences which do not make much sense. For example, on l134-135: "and the resulting posterior distribution on f as where" or l232: "By retaining the optimization-friendly propertie of convex problems ensured by to unimodality". These are just two examples, but there are many more and the authors should really focus on polishing the paper much more than it currently is.

- The figures/illustrations in this paper are of poor quality. It is hard to see what these figures are presenting without zooming very strongly. Even after doing so, it is hard to intepret these figures, and the captions do not tend to be particularly helpful. For example, in Figure 1, What does the coloring mean? Why is there no scale for what the coloring refers to? Why are the axes not named?

Comparison to existing methods:
- The discussion of computational cost is really minimal. Sure, the methods seem to perform well in terms of accuracy on the problem studied, but it is of course important to acknowledge that their cost will typically be much larger than for standard GPs. Currently, you are only providing cost for a simple one dimensional problem, which doesnt seem very representative. The paper could therefore clearly do with a detailed section discussing computational cost, and then a detailled assessment of the computational cost in the numerical experiments.

- The comparison and representation of Altamirano et al. 2024 is not always fair and at times even misleading.

For example, l52-53 on p1 states: "compared to the earlier work of Altamirano et al 2024 [...] added hyperparameters (\beta, c), did not support maximum likelihood estimation of hyperparameters [...]".

Firstly, the method of Altamirano et al works with a general weighting function, and the authors happen to parametrise such a weighting function with \beta and c and explain how to set these parameters. This is exactly what the present paper does with b and \epsilon (in fact b acts very similarly to \beta, whilst c acts very similarly to \epsilon). So it is misleading to imply that RCGP has additional hyperparameters compared to the proposed method.

Secondly, since RCGPs use generalised Bayes, using the maximum likelihood does not make sense (since the model is misspecified) and they therefore propose to use cross-validation, which is very widely used in the GP literature. This is shown to perform well. In the present paper, the authors propose to maximise the marginal likelihood for a model they know to be misspecified, which does not make much sense. Therefore saying that RCGP do not use the marginal likelihood and presenting this as a weakness without any evidence that it is seems unfair.

I am also wondering if the implementation of RCGP is done in a fair way. RCGPs are specifically well suited for GP regression where the Gaussian likelihood is mostly correct up to a small number of corrupted observations. This is clearly not the case in some of the settings that are studied in the experiments, which explains why RCGP may perform poorly. As discussed in the paper (l224), the authors expect a large number of extreme observations and therefore select b/\epsilon accordingly (this is certainly the case for the first experiment when setting). Does this mean you selected \beta/c for RCGPs accordingly to the recommendations in Altamirano et al to ensure that the methods are comparable? Sadly I was not able to verify this by looking at the code submitted. Also, which prior mean function was used? As discussed in the paper RCGPs perform poorly with a poor prior mean, so it would be interesting to see how the impact of this choice changes the behaviour of the method and whether this impacts how competitive it is.

- Theory: Theorem 1 is stated without a single assumption, but presumably you do indeed need assumptions for the result to hold. For example, what value of n does this hold for? Are the limitations on the various parameters needed for this to hold? For example, the inverse of the Gram matrix must be well-defined for v to be well-defined. What conditions on the kernel are needed for this? I would strongly recommend that this is formulated and proved more formally.

**Questions:**

- Can we really say that Huber-likelihood methods are preferable to alternatives in the settings you have studied? The answer of course also depends on computational cost. Therefore, please provide a full comparison of computational cost between all methods and for all experiments. Without this, it is not possible to make broad claims about HuberMCMC and HuberLA being preferable to existing methods. Please also provide details of your implementation and computational resources used. For example, how many iterations of gradient descent did you do for selecting hyperparameters of RCGP, and is this comparable to what you use for other methods?

- Have you considered using your standardised weights with other methods? Please provide an implementation of RCGP where the weights are selected according to your proposed standardised residuals instead of the weight proposed by Altamirano. This should be a straightforward modification of your existing code, and would allow us to check how much is gained by the Huber likelihood and how much is gained by the standardised residuals.

- Have you looked at how the selection of hyperparameters b and epsilon impact the performance of your method? Please provide full algorithmic details for every method you are using, as well as details of how hyperparameters were selected in each case. I would also strongly recommend a study of how this choice of hyperparameter impacts the performance of the method, and how this compares to competitors.

- Why is the last experiment only performed with Huber-based methods? I think it would make sense to implement all competitors on it to get a full picture of what is happening here.

- Is it possible to prove theoretically the robustness of the approach? If not, this should probably be mentioned as a drawback. But if so, then it would be worth adding this result.

---

> ### Comment · Reviewer_rnXe · 2024-11-26
>
> Since the discussion phase is ending today and I have not received a response to my concerns over the last 10 days, I will maintain my score.

---

> ### Author Response · Authors · 2024-11-26
> **Computational times**
>
> We thank the reviewer for their detailed comments and valuable suggestions. The discussion period is extended till Dec 2nd.
> Please find our comments below:
>
> > Overall, I found the paper to be poorly written. There are numerous times where there are clearly missing words or sentences which do not make much sense. For example, on l134-135: "and the resulting posterior distribution on f as where" or l232: "By retaining the optimization-friendly properties of convex problems ensured by to unimodality". These are just two examples, but there are many more and the authors should really focus on polishing the paper much more than it currently is.
>
> We have proofread the paper for missing words/ sentences. We'll share the revised version shortly.
>
> > The figures/illustrations in this paper are of poor quality.
>
> We are working on improving the clarity of labels and the quality of figures and will share them shortly.
>
> > The discussion of computational cost is really minimal. Sure, the methods seem to perform well in terms of accuracy on the problem studied, but it is of course important to acknowledge that their cost will typically be much larger than for standard GPs. Currently, you are only providing cost for a simple one dimensional problem, which doesn't seem very representative. The paper could therefore clearly do with a detailed section discussing computational cost, and then a detailed assessment of the computational cost in the numerical experiments.
>
> Below, we present a comparison of the computational cost for a single run on the Neal, Energy, and Yacht datasets.
>
> Execution Times (Seconds) for a Single Run on the **Neal Dataset**
>
> |    | SCtMCMC       | tLA           | HuberMCMC     | HuberLA       | RCGP         | GP           | LaplaceMCMC  |
> |--------|---------------|---------------|---------------|---------------|--------------|--------------|--------------|
> | Case 1 | 7.14 (0.31)   | 0.50 (0.14)   | 8.14 (0.38)   | 1.84 (0.47)   | 1.12 (0.01)  | 0.17 (0.04)  | 8.12 (0.37)  |
> | Case 2 | 7.32 (0.41)   | 0.69 (0.09)   | 8.37 (0.77)   | 1.64 (0.16)   | 1.23 (0.00)  | 0.19 (0.03)  | 8.34 (1.01)  |
> | Case 3 | 10.86 (1.20)  | 0.76 (0.27)   | 12.09 (0.41)  | 2.75 (0.25)   | 1.18 (0.00)  | 0.28 (0.01)  | 11.41 (0.45) |
> | Case 4 | 10.14 (1.13)  | 0.56 (0.22)   | 11.62 (0.31)  | 2.45 (0.26)   | 1.27 (0.00)  | 0.28 (0.02)  | 11.20 (0.33) |
>
>
>
> HuberLA—similar to RCGP and tLA—requires less computational time than HuberMCMC, as expected. The models converge faster for unidimensional data: HuberMCMC performs comparably to MCMC techniques with Student-t likelihood. For multidimensional cases, HuberMCMC takes longer to converge, whereas HuberLA converges more quickly.
>
> We will add these discussions in the revised paper.
>
> > The comparison and representation of Altamirano et al. 2024 is not always fair and at times even misleading. Firstly, the method of Altamirano et al works with a general weighting function, and the authors happen to parametrise such a weighting function with \beta and c and explain how to set these parameters. This is exactly what the present paper does with b and \epsilon (in fact b acts very similarly to \beta, whilst c acts very similarly to \varepsilon). So it is misleading to imply that RCGP has additional hyperparameters compared to the proposed method.
>
> We will correct this in the revised paper.
>
> >Saying that RCGP do not use the marginal likelihood and presenting this as a weakness without any evidence that it is seems unfair
>
> We will correct this in our paper and remove that sentence which said that.

---

> ### Author Response · Authors · 2024-11-26
>
> > I am also wondering if the implementation of RCGP is done in a fair way. RCGPs are specifically well suited for GP regression where the Gaussian likelihood is mostly correct up to a small number of corrupted observations. This is clearly not the case in some of the settings that are studied in the experiments, which explains why RCGP may perform poorly. As discussed in the paper (l224), the authors expect a large number of extreme observations and therefore select b/\epsilon accordingly (this is certainly the case for the first experiment when setting). Does this mean you selected \beta/c for RCGPs accordingly to the recommendations in Altamirano et al to ensure that the methods are comparable?
>
> As suggested by Altamirano et al., we used the inverse multi-quadratic weighting function for RCGP. The implementation is shown in the code snippet below:
> ```
>  standard_gp = gpflow.models.GPR(
>     (x, y),
>     kernel=gpflow.kernels.SquaredExponential(lengthscales=0.1,variance = 0.05),
>     noise_variance = 0.01
> )
> ```
> We trained a standard GP model first to initialize the noise variance for RCGP:
> ```
>  model_rcgpr = RCGPR(
>             (x, y),
>             kernel=gpflow.kernels.SquaredExponential(),
>             weighting_function=IMQ(1.5),
>             noise_variance=standard_gp.likelihood.variance  # Example noise variance
>         )
>
> ```
>
> We chose a GP prior with zero mean function and squared exponential kernel for the RCGP, to maintain consistency across all baseline methods.
>
> We ran our experiments on a Windows system with  Intel(R) Core(TM) i7-1260P on 32 GB RAM.
>
> > Have you considered using your standardised weights with other methods?
>
> Yes, Tables 1 and 2 in the rebuttal.pdf show results with pursuit weighting applied to all baselines (see response to reviewer UKZY).
>
> > Please provide an implementation of RCGP where the weights are selected according to your proposed standardised residuals instead of the weight proposed by Altamirano. This should be a straightforward modification of your existing code, and would allow us to check how much is gained by the Huber likelihood and how much is gained by the standardised residuals.
>
>
> Implementation of pursuit weights in RCGP:
> we define a new class RCGPWPW whose leave-one-out cross-validation function is modified to scale the residuals:
> ```
> def loo_cv(self) -> tf.Tensor:
>         """
>         Computes the leave-one-out cross-validation (LOO-CV) loss for training.
>         Adjusts residuals using pursuit weights (pw).
>         """
>         X, Y = self.data
>         mean_output = tf.cast(self.mean_function(X), Y.dtype)
>         err = (Y - mean_output) / self.pw  # Adjust residuals with pursuit weights
>         K = self.kernel(X)
>         n = tf.cast(tf.shape(X)[0], K.dtype)
>         likelihood_variance = self.likelihood.variance_at(X)
>         W, W_dy = self.weighting_function.w_dy(X, err)
>         dylog2 = 2 * likelihood_variance * W_dy / W
>         Y_bar = err - dylog2
>
>         K_sW = add_noise_cov(K, tf.squeeze(W, axis=-1), tf.squeeze(likelihood_variance, axis=-1))
>         L_sW = tf.linalg.cholesky(K_sW)
>         L_sW_inv = tf.linalg.inv(L_sW)
>         diag_K_sW_inv = tf.reshape(tf.reduce_sum(L_sW_inv ** 2, axis=0), (-1, 1))
>
>         A = diag_K_sW_inv * dylog2
>         B = tf.matmul(L_sW_inv, tf.matmul(L_sW_inv, Y_bar), transpose_a=True)
>         C = diag_K_sW_inv * (1 - diag_K_sW_inv * (likelihood_variance * (W**-2) - likelihood_variance))
>         D = C / diag_K_sW_inv**2
>
>         loo = -0.5 * tf.reduce_sum(tf.math.log(D))
>         loo -= 0.5 * n * np.log(2 * np.pi)
>         loo -= 0.5 * tf.reduce_sum((A + B) ** 2 / C)
>         return loo
> ```
>
> with pursuit weights:
>
> ```
> def calculate_pursuit_weights(self, X, cutoff_scale=0.975):
>         """
>         Calculate pursuit weights for TensorFlow tensors based on projection statistics.
>
>         Args:
>             X: A TensorFlow tensor of shape [m, n].
>             cutoff_scale: Cutoff scale for chi-squared distribution.
>
>         Returns:
>             pw: A TensorFlow tensor of shape [m, 1] containing pursuit weights.
>         """
>         X = tf.cast(X, tf.float64)  # Ensure X is float32
>         ps = self.projectionstatistics(X)  # Compute projection statistics
>         cutoff = chi2.ppf(cutoff_scale, df=int(X.shape[1]))  # Chi-squared cutoff
>         cutoff = tf.constant(cutoff, dtype=tf.float64)  # Ensure dtype compatibility
>         pw = tf.minimum(1.0, cutoff / tf.square(ps))  # Calculate weights
>         # print("PW:", pw)
>         return tf.expand_dims(pw, axis=-1)
> ```
> The full code is available at rcgpwPW @https://anonymous.4open.science/r/GpHuber-6A2D.
>
> Tables 1 and 2 in the rebuttal.pdf compare performances with pursuit weights added across all baselines.
>
> We observed that adding pursuit weights significantly improved the performance of LaplaceMCMC and GP. However, models using Student's-t likelihood and RCGP showed only marginal improvements.

---

> ### Author Response · Authors · 2024-11-26
>
> > Have you looked at how the selection of hyperparameters b and epsilon impact the performance of your method? Please provide full algorithmic details for every method you are using, as well as details of how hyperparameters were selected in each case. I would also strongly recommend a study of how this choice of hyperparameter impacts the performance of the method, and how this compares to competitors
>
>
> We conducted experiments on the Neal dataset with $b$ varying from $1$ to $2.5$ in $0.1$ increments, while setting $\varepsilon = 0.45$ for the Student-t noise case. Figure 2 (a) and (c) in the rebuttal.pdf show the RMSE and MAE results. Next, we fixed $b = 1.5$ and plotted RMSE and MAE in Figures 2 (b) and (d) for $\varepsilon$ varying between $0.1$ and $0.9$.
>
>
> We aim to make GP-Huber to be robust for contamination $\varepsilon = 0.45$. We noticed optimal setting to be: $b = 1.5$ for non-Gaussian noises and $b = 0.5$ for Gaussian noise settings through our experimentation. In the Student's-t noise setting, we observed that increasing $\varepsilon$ beyond 0.45 led to higher RMSE and MAE, reducing overall performance. Similarly, increasing the threshold $b$ while keeping $\varepsilon = 0.45$ also resulted in decreased performance.
>
> Details of the other baselines:
>
> | Model #      | Description                                                                                                                                                  |
> |--------------|--------------------------------------------------------------------------------------------------------------------------------------------------------------|
> | **SCtMCMC**  | GP with a Student's t error model using a scale mixture representation, solved via MCMC integration of the latent vector $\mathbf{f}$, likelihood hyperparameters ${\phi}=(\nu,\sigma^{2})$, and kernel hyperparameters ${\theta}$. |
> | **tLA**      | Student's-t likelihood model with the Laplace approximation, solved via Laplace integration over $\mathbf{f}$ and MAP estimates of ${\phi}=(\nu,\sigma^{2})$ and ${\theta}$. |
> | **GP**       | Conjugate model: GP with normal error, where the hyperparameters include $\mathbf{f}$, ${\phi}=\sigma^{2}$, and ${\theta}$. |
> | **LaplaceMCMC** | GP with Laplace likelihood, solved via MCMC integration over $\mathbf{f}$, ${\phi}=\sigma$, and ${\theta}$.                                           |
>
>
> > Why is the last experiment only performed with Huber-based methods? I think it would make sense to implement all competitors on it to get a full picture of what is happening here.
>
> We compared GP-Huber with the state-of-the-art method for transmission spectroscopy within GP framework proposed by Gibson et. al. to model instrumental systematics. Unlike conventional regression tasks mapping $x \to y$, this problem here is to estimate the planet-to-star radius ratio, $\rho_{\text{radius}}$, a parameter in $\mathbf{\phi}$, of $T(\mathbf{x}, \phi)$, where $T$ represents a quadratic limb-darkening function. Implementing this with Student's-t or Laplace likelihoods in GP models is not trivial and the Python packages utilized by Gibson et al. are now deprecated.
>
> For real-world applications, we have have another comparison on the Twitter flash crash dataset.
>
> >  Is it possible to prove theoretically the robustness of the approach? If not, this should probably be mentioned as a drawback. But if so, then it would be worth adding this result.
>
> Yes, we prove the bounded influence of observations on the Huber likelihood which ensures robustness in Theorem 2:
>
> *Under the same assumptions as Theorem 1, the influence of an individual observation $y_i$ on the posterior mean $\mathbb{E}[f(x) \mid \mathbf{y}]$ is bounded, ensuring robustness to outliers:*
> $\left|\frac{\partial}{\partial y_i} \mathbb{E}[f  \mid {y}]\right| \leq \frac{b}{\sigma}$.
> See, rebuttal.pdf for proof.

---

> ### Author Response · Authors · 2024-11-26
> **GP-Huber theoretical robustness**
>
> > Theory: Theorem 1 is stated without a single assumption, but presumably you do indeed need assumptions for the result to hold. For example, what value of n does this hold for? Are the limitations on the various parameters needed for this to hold? For example, the inverse of the Gram matrix must be well-defined for v to be well-defined. What conditions on the kernel are needed for this? I would strongly recommend that this is formulated and proved more formally.
>
> We updated the statement of the Theorem 1 to:
>
> *Let $\mathcal{D}$
> be a dataset with distinct covariates
> $x_{i} \in \mathcal{X}$ and response $y_{i} \in \mathcal{Y}$, where $n < \infty$. The kernel matrix $K \in \mathbb{R}^{n \times n}$ is positive definite, with elements $K_{ij} = k(x_{i}, x_{j})$ defined by a continuous kernel function $k: \mathcal{X} \times \mathcal{X} \to \mathbb{R}$.  Assume the Huber likelihood function $p_{H}(y| f, \sigma)$ is based on a strictly convex and continuous Huber loss $\rho(r_{i}): \mathbb{R} \to \mathbb{R}$. Then, the posterior distribution $p(f | D, \theta, \sigma)$ is unimodal.*
>
> For the revised proof, please refer to the rebuttal.pdf @https://anonymous.4open.science/r/GpHuber-6A2D/rebuttal.pdf.
>
> > Can we really say that Huber-likelihood methods are preferable to alternatives in the settings you have studied? The answer of course also depends on computational cost. Therefore, please provide a full comparison of computational cost between all methods and for all experiments. Without this, it is not possible to make broad claims about HuberMCMC and HuberLA being preferable to existing methods. Please also provide details of your implementation and computational resources used. For example, how many iterations of gradient descent did you do for selecting hyperparameters of RCGP, and is this comparable to what you use for other methods?
>
> We presented the computational cost for all baselines on the Neal dataset across various outlier and noise cases in the table above. For the Yacht and Energy datasets, performance values were adopted from Altaminaro's paper to compare with HuberLA and HuberMCMC.
> HuberLA: demonstrated convergence rates comparable to RCGP.
> HuberMCMC: required more time as a Monte Carlo method; it converged faster for unidimensional covariate cases but typically needed 20–30 seconds in multidimensional cases.

---

### Official Review · Reviewer_dhjR · 2024-10-31

**Soundness:** 3
**Presentation:** 4
**Contribution:** 3
**Rating:** 6
**Confidence:** 4

**Summary:**

The paper introduces a Gaussian Process regression that is robust towards extreme output outliers. The GP is based on a Huber likelihood leading to GP-Huber posterior distribution that is estimated using Laplace approximation. The authors also propose to address the input  outliers by introducing a projection pursuit weights that alleviates their influence on the model performance. The efficiency of the method is demonstrated on simulated data and real-world data, compared to the Robust Gaussian Process introduced by Altamirano et al (2024). The authors also provide some numerical results on transmission spectroscopy experiments.

**Strengths:**

Overall, the paper reads well and is easy to follow. The paper makes some good contributions to the field of robust GP regression research.

- The authors propose to use a Huber likelihood which leads to a GP-Huber posterior distribution. Theorem 1 ensures that the GP-Huber posterior is unimodal, which enables to perform inference and hyperparameter optimization. Using Huber loss to address robustness is known, but its use for GP regression seems to be new.

- The authors also introduce some weight pursuit to address the issue of having both corrupted output and corrupted input. The method seems to be efficient and easy to implement.

- The authors propose to use the Laplace approximation and MCMC approximation to compute the intractable integral. This enables to compute predictive points and to optimize the marginal likelihood. Although this approach is standard in variational inference for GP regressions, the paper addresses the subtleties that come from the choice of Huber density. In particular, they use the pseudo-Huber loss to ensure continuity of the derivatives.

- The numerical experiments are well explained and they show the efficiency of the approach. Especially, the transmission spectroscopy experiment shows the relevance of the approach in a specific context.

**Weaknesses:**

Although I like the paper, I find that the paper presents some weaknesses.

- Although it looks like Huber likelihood has not been used for GP regression, the approach combines a number of known variational inference tools (MCMC and Laplace approximation). Here I am not questioning the relevance of the approach but rather I am just pointing out that the main results of the paper are somewhat expected.

- A naive comment: It seems that the approach performs better than the state-of-the-art on the extreme outlier cases. (Case 2) However, the performance does not seem better for milder outlier cases. (Case 4). It looks like the model is performing well on the extreme cases. However, in practice, these extreme outliers are often removed from the training set through a preprocessing step (using some basic thresholding). Could the authors discuss about the main differences between "extreme cases" versus "easier cases"?

- Of course, extreme outliers is an important issue and the presented model is highly relevant to treat such cases. However, identifying outliers is more challenging when the variance $\sigma^2$ increases and/or when the variance is heteroscedastic with some high variations areas. This would be interesting to observe the behavior of the model in such cases. A subsequent question is: Would it be possible to extend such model to heteroscedastic variance? Maybe the authors could discuss potential challenges that might raise from such a model extension?

- The choice of $\varepsilon$ and $b$ seems to be crucial and I like the paragraph explaining their impact. However, this leads to a naive question from a practitioner point of view. How would we know how to set such parameters? How does a conservative choice of $\varepsilon$ and $b$ impact the model efficiency when the presence of outliers is overestimated? The aim of this question is not to criticize the issues addressed by the model but rather to question the practical limitations. How should a practitioner proceed to set such parameters?

Minor comments:

- In first rows of Table 1, the numbers in bold are not the right ones.
- It would have been interesting to see the posterior mean results of RCGP in Figures 2,5, 6 and 7. This would help to understand where this model fails in the mean reconstruction.
- I encourage the authors to provide a bit more details about the setting of experiments in Section 6.1 and Section 6.2.

**Questions:**

See the first questions in the previous section.

- It seems that HuberMCMC performs poorly on the Energy dataset (Section 6.2) for focused outliers. How would the authors explain it?

- How do the authors generate cases 3 and 4 compared to cases 1 and 2? It seems that the output data are upper bounded in Figures 6 and 7.

---

> ### Author Response · Authors · 2024-11-25
>
> Thank you for your thoughtful feedback and for acknowledging the relevance and contribution of our approach.
>
> > Although it looks like Huber likelihood has not been used for GP regression, the approach combines a number of known variational inference tools (MCMC and Laplace approximation). Here I am not questioning the relevance of the approach but rather I am just pointing out that the main results of the paper are somewhat expected.
>
> We understand the importance of bringing novelty to inference methods, especially with so many advanced variational techniques available. However, in the GP-Huber model, we establish:
>
> **Theorem 1**   *The GP-Huber posterior distribution  $f| \mathcal{D}, \theta, \sigma$  is unimodal.*
>
> Thus, by ensuring derivative continuity, we chose simpler methods like Laplace approximation and Gibbs sampling.
>
> We understand that combining known variational inference tools like MCMC and Laplace approximation might suggest that the results are expected. However, we would like to highlight several key aspects of our work that we believe are novel and non-trivial:
> -  Inference methods using the Huber loss as a likelihood in Gaussian Process regression have, to our knowledge, not been proposed before.
> - Our approach is the first within a GP framework to propose a mechanism that simultaneously manages outliers in both the covariate and response dimensions, including handling multivariate covariate outliers using projection statistical distances.
>
> Furthermore, we demonstrate that the bounded influence of observations theoretically guarantees the robustness of GP-Huber (see proof in rebuttal.pdf @https://anonymous.4open.science/r/GpHuber-6A2D/rebuttal.pdf).
>
> **Theorem 2**  *Under the same assumptions as Theorem 1 the influence of an individual observation $y_i$ on the posterior mean $\mathbb{E}[f \mid {y}]$ is bounded.*
>
> $\left|\frac{\partial}{\partial y} \mathbb{E}[f \mid {y}]\right| \leq \frac{b}{\sigma}$.
>
> >  It seems that the approach performs better than the state-of-the-art on the extreme outlier cases. (Case 2) However, the performance does not seem better for milder outlier cases. (Case 4). It looks like the model is performing well on the extreme cases. However, in practice, these extreme outliers are often removed from the training set through a preprocessing step (using some basic thresholding).
>
> Removing extreme outliers through thresholding may seem convenient, but it risks discarding valuable insights, leading to biased models and reduced generalizability.
>
> > Could the authors discuss about the main differences between "extreme cases" versus "easier cases"?
>
> Extreme cases include outliers $y^{(l)}$ that are focused far away from the main data cloud (similar to focused outliers in Altaminaro's cases; see Figure 1(b) in the rebuttal.pdf). The main difference between extreme and near outliers lies in their magnitude and pattern. Near outliers $y^{(c)}$, seen in cases 3 and 4, are scattered but closer to the main data in response dimensions (similar to the asymmetric case in Altaminaro et al.'s paper). In cases 1 and  3, along with $y^{(l)}$ and $y^{(c)}$, we introduced outliers in the multiple covariate dimensions ${x}^{(l)}$. To further illustrate, we add a new results in the table below where $y^{(l)}$ and $y^{(c)}$ are introduced with ${x}^{(l)}$, highlighting GP-Huber's robustness to extreme as well as near outliers in response dimensions along with covariate outliers. These are the outlier occurrences where most existing models in the literature struggle significantly. We ran 10 experiments with random Gaussian noise addition and hyperparameter initializations. The table below reports the mean RMSE and MAE, with standard deviations in brackets.
>
>   | Metric | SCMCMC | tLA | HuberMCMC  | HuberLA | RCGP | GP  | LaplaceMCMC|
> |--------|-------------|----------|----------------|--------------|-----------|---------|------------------|
> | RMSE   | 0.98 (1.00) | 0.81 (0.11) | **0.55 (0.04)**   | 0.75 (0.02)  | 3.11(0.07) | 1.73 (0.01) | 0.62 (0.04)       |
> | MAE    | 0.55 (0.33) | 0.52 (0.06) | **0.42 (0.03)**   | 0.54 (0.02)  | 2.37 (0.03) | 1.39 (0.01) | 0.45 (0.02)       |

---

> ### Author Response · Authors · 2024-11-25
> **Heteroscedastic GP-Huber**
>
> > Of course, extreme outliers is an important issue and the presented model is highly relevant to treat such cases. However, identifying outliers is more challenging when the variance  increases and/or when the variance is heteroscedastic with some high variations areas. This would be interesting to observe the behavior of the model in such cases. A subsequent question is: Would it be possible to extend such model to heteroscedastic variance? Maybe the authors could discuss potential challenges that might raise from such a model extension?
>
> Yes, we extended GP-Huber to handle heteroscedastic noise: $y_{i}=f(x_{i}) +e_{i}$ where $e_{i} \sim \mathcal{N}(0, \sigma^{2}(x_{i}))$. We lay a GP-Huber prior on the main latent variables of $f$ of y-process $f_{y}\sim \textrm{GP-Huber}(0, k_{y}(x_{i}, x_{j}))$. Next, we define another GP-Huber for learning noise variances $t= \log(\textrm{Var}(y-f(x)))$, $f_t \sim \textrm{GP-Huber} (0, k_{t}(x_{i}, x_{j}))$. We would be happy to include this as a comment in our paper if you recommend it and hope that this will allow you to fully recommend our work for acceptance.
>
>
> We compared our HuberMCMC inference (H-HuberMCMC) with VHGP (Variational Heteroscedastic Gaussian Process) [1] and NSGP (Non-Stationary Gaussian Process with noise variance as the only non-stationary parameter) [2] on Silverman's motorcycle dataset. The table below shows the results from 10 random experiments with an 80% train-test split, reporting the mean and standard deviation (in brackets).
>
> | Metric    | H-HuberMCMC | VHGP   | NSGP   |
> |-----------|-------------|--------|--------|
> | RMSE      |     0.1068 (0.006) |     0.1036 (0.002) |     0.1034 (0.002) |
> | MAE       | 0.081 (0.0066)      | 0.0766 (0.001) | 0.0762 (0.001) |
> | NLPD      | 1.0022  (0.00)   | -1.2777 (0.00) | -1.1378 (0.00)
>
>
> The initial implementation of the heteroscedastic GP-Huber, inspired by the most likely method by Kersting et al. [3], shows performance comparable to baselines like VHGP and NSGP. In the future, we aim to develop a more refined and robust implementation of the heteroscedastic GP-Huber.
>
>
> *PS: With the limited time, we were only able to implement the HuberMCMC inference within GP-Huber.*
>
> References:
>
> [1] Lázaro-Gredilla, Miguel, and Michalis K. Titsias. "Variational Heteroscedastic Gaussian Process Regression." ICML. 2011.
>
> [2] Heinonen, Markus, et al. "Non-stationary gaussian process regression with hamiltonian monte carlo." Artificial Intelligence and Statistics. PMLR, 2016.
>
> [3] Kersting, K., Plagemann, C., Pfaff, P., & Burgard, W. (2007, June). Most likely heteroscedastic Gaussian process regression. In Proceedings of the 24th international conference on Machine learning (pp. 393-400).
>
>
> > The choice of $b$ and  seems to be crucial and I like the paragraph explaining their impact. However, this leads to a naive question from a practitioner point of view. How would we know how to set such parameters? How does a conservative choice of
>  and impact the model efficiency when the presence of outliers is overestimated? The aim of this question is not to criticize the issues addressed by the model but rather to question the practical limitations. How should a practitioner proceed to set such parameters?
>
> We solve $\epsilon$ for each $b$ by solving the equation \( 2\Phi(b) - 1 + \frac{2}{b} \phi(b) = \frac{1}{1 - \epsilon} \) (Equation 36 in the paper). The values are listed in the table below (also plotted in Figure 1 (a) in rebuttal.pdf) ) :
>
> | $\epsilon$ | $b$   |
> |------------|-------|
> | 0.10       | 3.16  |
> | 0.20       | 2.24  |
> | 0.30       | 1.83  |
> | 0.40       | 1.58  |
> | 0.50       | 1.41  |
> | 0.60       | 1.29  |
> | 0.70       | 1.20  |
> | 0.80       | 1.12  |
> | 0.90       | 1.05  |
> | 1.00       | 1.00  |
>
>
> > Minor comments
>
> We’ve revised our paper with corrected bolding in Table 1. We will add RCGP figures (Figures 2, 5, 6, 7), and expanded experimental details. We will share the revised version shortly and are open to incorporating any further suggestions you may have.
>
> > It seems that HuberMCMC performs poorly on the Energy dataset (Section 6.2) for focused outliers. How would the authors explain it?
>
> HuberMCMC excels with extreme outliers, such as in cases 1 and 2 (see paragraph on line 394 of the paper), where magnitudes far exceed the maximum response value. In contrast, on Altaminaro et al.'s Energy dataset with focused outliers closer to the response data, HuberLA performed better, as expected.
>
>
> > How do the authors generate cases 3 and 4 compared to cases 1 and 2? It seems that the output data are upper bounded in Figures 6 and 7.
>
> Outliers were added randomly: $x^{(l)}$ in covariates and $y^{(l)}, y^{(c)}$ in responses. Extreme outliers $y^{(l)}$ in cases 1 and 2  were placed at least 3 standard deviations beyond the maximum response, while near outliers $y^{(c)}$ in cases 3 and 4 were within 2 standard deviations. The locations of $x^{(l)}$ may or may not overlap with $y^{(l)}$ or $y^{(c)}$.

---

> ### Author Response · Authors · 2024-11-25
> **Code details**
>
> Finally, we present the implementation details in MATLAB of our attempt to implement heteroscedastic GPs within the limited time we had, based on the most likely heteroscedastic Gaussian process method proposed by Kersting et al.
>
>
> ```
>  gpcf = gpcf_sexp('lengthScale', 1, 'magnSigma2', 0.05, ...
>                  'lengthScale_prior', pl, 'magnSigma2_prior', pm);
> % Create the likelihood structure
> lik = lik_huber('sigma2', 0.1^2, 'sigma2_prior', pn, 'weights', weights);
>
> % ... Finally create the GP structure
> gp_f = gp_set('lik', lik, 'cf', gpcf, 'jitterSigma2', 1e-9, ...
>               'latent_method', 'MCMC', 'mode', 'homo');    % setting the mode as homoscedastic
> gp_t = gp_set('lik', lik, 'cf', gpcf, 'jitterSigma2', 1e-9, ...
>               'latent_method', 'MCMC');
> gp_ft = gp_set('lik', lik, 'cf', gpcf, 'jitterSigma2', 1e-9, ...
>                'latent_method', 'MCMC', 'mode', 'hetero'); % setting the mode as heteroscedastic
>
> [f_gp, var_f] = fit_homoscedastic_gp(gp_f, x, y);
> z = log(var_f);
> % let us define z = log(sigma)
>
> for iter = 1:max_iter
>     %  (training f-process)
>     [Ef0, Varf0] = gp_pred(f_gp, x, y);
>
>     % (training t-process)
>     z_gp = fit_homoscedastic_gp(gp_t, x, z);
>
>     %
>     [z, ~] = gp_pred(z_gp, x, z, x);  % predict considering rr
>
>     [ft_gp] = fit_heteroscedastic_gp(gp_ft, x, y);
>
>     [Ef, Varf] = gp_pred_hetero(ft_gp, x, y, x, exp(z), exp(z));  % gp_pred_hetero(gp, input, output, test input, var_train, var_test)
>
>     % Check for convergence
>     [converged, values] = has_converged(Ef0, Varf0, Ef, Varf, tol);
>     VAL(iter, :) = values;
>
>     if converged
>         break;
>     else
>         gp_f = gp_ft;
>         gp_f.mode = 'homo';  % setting the mode as homoscedastic
>         [f_gp, var_f] = fit_homoscedastic_gp(gp_f, x, y);
>         z = log(var_f);
>     end
> end
> ```

---

> ### Comment · Reviewer_dhjR · 2024-11-26
>
> I thank the authors for this detailed answers. I find that most of the answers are convincing.
> However, I still have concerns about the usage of such an approach in practice from high level perspective. The extreme outliers case seems to be easily identified using some data analysis and can be treated with basic methods.
> Overall I find the paper good and I decide to keep my score.

---

> ### Author Response · Authors · 2024-11-26
>
> Thank you! The proposed GP-Huber can handle extreme as well as various combinations of extreme and near outliers (even near outliers solely very well). As reviewer rnXe highlighted, a key strength of our method is its ability to address outliers in both the covariate and response dimensions. We believe that among all the presented outlier cases (only on Neal dataset) it is possible to identify outliers in cases 2 and 4 through data analysis. As a real-world application, we demonstrated our method’s effectiveness in estimating the planet-to-star radius ratio. We used noisy flux measurements from orbit 1, which are often discarded in state-of-the-art methods. Such noisy measurements arise through complex instrumental correlations and cannot be handled using basic data analysis. We considered focused and symmetric outlier cases in the Energy and Yacht datasets—presented in the study by Altaminaro et al. HuberLA and HuberMCMC show good performance values. Similarly, on the Twitter Flash Crash dataset: comparisons highlight the advantages of the proposed method.
>
>  Additionally, we would like to clarify that Figure 1(b) in the rebuttal is only an illustration of outlier cases. Outliers appear randomly, and in Case 5, we added outliers arbitrarily, including some closer to the data cloud. The 2 and 3 standard deviation thresholds mentioned above are a general guide, but the outliers were placed randomly.
>
> The added outliers were also masked i.e. at  many locations of $x^{(l)}$ we also encounter $y^{(c)}$ (in Case 3) or $y^{(l)}$ (in Case 4). It is not possible to track down such masked outliers with basic thresholding methods. From practical implementation view, the method is easy to implement and can be extended to the sparse cases.
>
> It is clear in the table below, the GP-Huber has advantage in dealing with the outliers in the covariates and response dimensions simultaneously.
>
>  # Results on Neal dataset with outliers $x^{(l)}, y^{(l)}$, and $y^{(c)}$
>  | Metric | SCMCMC | tLA | HuberMCMC  | HuberLA | RCGP | GP  | LaplaceMCMC|
> |--------|-------------|----------|----------------|--------------|-----------|---------|------------------|
> | RMSE   | 0.98 (1.00) | 0.81 (0.11) | **0.55 (0.04)**   | 0.75 (0.02)  | 3.11(0.07) | 1.73 (0.01) | 0.62 (0.04)       |
> | MAE    | 0.55 (0.33) | 0.52 (0.06) | **0.42 (0.03)**   | 0.54 (0.02)  | 2.37 (0.03) | 1.39 (0.01) | 0.45 (0.02)       |
>
>
> With the heteroscedastic version, we show that the method can be extended to handle input dependent outliers.
>
>
> ## An ask: since the discussion period goes on for more 6 days, would you like us to conduct additional experiments or provide any clarifications to support our paper?

---

### Official Review · Reviewer_UKZY · 2024-10-31

**Soundness:** 3
**Presentation:** 3
**Contribution:** 2
**Rating:** 5
**Confidence:** 5

**Summary:**

The paper proposes a robust Gaussian process regression model in covariates and outputs. For the output outliers, the authors propose to use the Huber likelihood, reducing deviations caused by outliers in output data. This likelihood yields a non-conjugate posterior, so the authors propose to use two standard ways to deal with intractable posteriors: Laplace approximation and Gibbs sampling. The paper employs a distance-based down-weighting scheme for outliers in the input space.

**Strengths:**

Input and output outliers are a relevant problem in the GP community, and I think the method could have a real impact on that.

**Weaknesses:**

I find it challenging to identify the main contribution of this paper. On the one hand, it presents a method for handling output outliers using the Huber likelihood. On the other, it proposes a process to guard against covariate outliers.

However, the authors treat both methods as a single approach in the experiment section, which is surprising since the projection technique could be applied to other GP models to make them similarly robust to covariate outliers. This treatment blurs the paper’s contribution and makes it challenging to follow. I would recommend distinguishing between the two methods more clearly, as this could clarify each contribution's unique impact and utility.

**Questions:**

- I find Figure 1 not easy to understand. I don't see the real benefits of using a 3D example here since the position of a 2D example will be much clearer.

- The authors claim that 'Our approach does not introduce Huber likelihood specific parameters in the posterior inference,
avoiding the need for additional model-specific tuning'. However, I think this is somehow misleading since the model introduces two extra parameters $\beta$ and $\epsilon$. While the authors suggest a heuristic to set both parameters (which assumes that you want to guard against 10% outliers ), the method doesn't indeed add extra parameters.

- The comparison in Table 1 seems unfair, as none of the methods compared account for covariate outliers. I believe this is the primary reason the Huber methods outperform the others. This becomes more evident in Tables 2 and 4, where it's unclear whether the proposed method consistently outperforms the rest.

- Similarly, if the authors propose a method to handle covariate outliers, it should be compared with other methods that address the same issue.

---

> ### Author Response · Authors · 2024-11-25
> **Additional experiments with pursuit weights added to the baselines**
>
> We thank the reviewer for their constructive feedback and valuable insights.
>
> >  I find it challenging to identify the main contribution of this paper.
>
>  Our main contribution is a unified approach to handle outliers in both covariate and response dimensions using Huber likelihood in GP framework.
> Although pursuit weighting may appear to address only covariate outliers—since it is computed on covariates (this step identifies outliers across multiple covariate dimensions)—it serves a broader purpose: scaling residuals iteratively until convergence (rather than a one-time pre-inference step).
> This determines the switching treatment between $\mathcal{l}_1$ and $\mathcal{l}_2$ loss. To preserve the key switching property of the Huber loss-based density function, we propose two inference methods—HuberMCMC and HuberLA—that address the unique challenges of integrating GP priors with the Huber likelihood. GP-Huber model, to our knowledge, the first such approach that handles response and covariate outliers.
>
> > I don't see the real benefits of using a 3D example in Fig 1 here since the position of a 2D example will be much clearer.
>
> We aimed to demonstrate the effects of outliers in a single dimension, $x_{1}$, within $[x_{1}, x_{2}]$, on the fitting curve. We are currently working on improving the figure's clarity and will share the updated version soon.
>
> >  'Our approach does not introduce Huber likelihood specific parameters in the posterior inference, avoiding the need for additional model-specific tuning.'
>
> We agree that the hyperparameters, $b$ and $\varepsilon$, of the Huber likelihood are specified using heuristics. We will add this in the paper.
>
> > The comparison in Table 1 seems unfair, as none of the methods compared account for covariate outliers. I believe this is the primary reason the Huber methods outperform the others. This becomes more evident in Tables 2 and 4, where it's unclear whether the proposed method consistently outperforms the rest.
>
> In cases 2 and 4, we excluded pursuit weighting (pw) in GP-Huber ensuring a fair comparison across baselines that can handle response outliers. As noted earlier, our method excels in cases 1 and 3, where outliers are present in both covariate and response dimension. We don't expect our method to work well in all the outlier cases. Tables 2 and 4 (in rebuttal.pdf) show that GP-Huber delivers performance comparable to other baselines even with  near outliers $y^{(c)}$.
>
> To ensure a fairer comparison, we incorporated pursuit weights into the baselines (denoted by +pw) and evaluated the results across all four noise scenarios and all outlier cases. Due to space constraints, we encourage you to refer to Tables 1, 2, 3, and 4 in the rebuttal.pdf available @https://anonymous.4open.science/r/GpHuber-6A2D/rebuttal.pdf . We also excluded projection weighting from the baselines (denoted as -pw) for cases 2 and 4, where $x^{l}$ was absent from the covariate dimensions.
> The experiments were conducted over 10 random simulations, with the mean and standard deviation of RMSE and MAE reported.
>
> >Similarly, if the authors propose a method to handle covariate outliers, it should be compared with other methods that address the same issue.
>
> We address outliers in both covariate and response dimensions, as outlined earlier. If there’s a specific covariate method you’d like us to compare against, we’re happy to evaluate it. To ensure fairness, we incorporated pursuit weighting into the baselines and assessed their performance (Tables 1 and 3, rebuttal.pdf): HuberMCMC outperformed baselines in cases 1 and 2, while HuberLA excelled in cases 3 and 4. Pursuit weighting improved accuracy for Laplace inference methods and Gaussian processes but reduced efficiency for Student-t likelihoods—consistent with the Huber likelihood’s Gaussian and Laplacian nature.
>
>
> *With these additional results, we hope the paper now adequately addresses the key issue of fairness in comparison.*

---

### Official Review · Reviewer_b8uR · 2024-11-03

**Soundness:** 2
**Presentation:** 2
**Contribution:** 2
**Rating:** 5
**Confidence:** 3

**Summary:**

This paper presents a novel method for robust gaussian process regression by using huber likelihood, which can deal with outliers in both covariate and output spaces. For covariate outliers, the paper introduces a projection pursuit weights to adjust their influence. For output outliers, the paper uses the huber likelihood. The posterior distribution, while intractable, is shown to be unimodal. Two approximate inference methods are proposed for this model, one based on MCMC and another based on Laplace approximation. Both methods are tested on a range of synthetic datasets and a real-world application on transmission spectroscopy, demonstrating their robustness and effectiveness.

**Strengths:**

- The proposed method is well-motivated to address previous approaches' sensitivity to heavy outliers.

- The choices of Huber likelihood and

- The presented model and inference procedure are easy to follow.

- the proposed Laplace approximation (LA) inference method seems to have favorable empirical and runtime performance.

**Weaknesses:**

The biggest advantage of the paper is that the advantage of the proposed approach is unclear.

First, while the use of huber likelihood is natural, there seems be a lack of discussions on the theoretical properties of Huber likelihood when compared to other robust likelihoods, e.g. student t, and their implications in Gaussian processes framework. Making this point clear would make the paper's contribution more significant other than just combining Gaussian process with any existing robust likelihood model.

On the empirical side, more thorough experimental details need to be presented. For each of the table presented, how many random simulations are run and what are the standard errors? See more comments on empirics in the questions below.

Finally, it would be great of have a -brief introduction of the RCGP approach, which is served as an important baseline in the paper.

**Questions:**

- In the motivating figure 1, what is the result from your proposed approach? Also can you plot the true sinc function?

- the paper claims that no additional hyperparameters need to be chosen. But there are indeed threshold parameters $b$ and contamination parameters $epsilon$ required for the Huber model. The last paragraph in section 4 discusses  one particular configuration, is this what the authors meant by eliminating the need for setting them?

- Experiment 6.1, can you elaborate the difference between Case 1 v.s. Case 3 (or Case 4)? Is is just that in case 3 & 4, outliers in y space are closer to the main cluster of the data?

- In table 1 (and other tables), no standard errors are reported. Also, in the first row of Tbale 1, Huber LA with RMSE 0.25 is better than Huber MCMC with RMSE 0.37, but why the latter is bolded. Same question regarding the first MAE row. Please go through the table to make sure the bolded numbers are indeed the best performance.

- Figure 2: where is the result for HuberMCMC? Also it would be nice to have a more concrete description on each method's qualitative performance.

- Figure 3.a: How are the mean function and GP-Huber curves obtained?

- Figure 3.b: Describe the red bands and the dots.

- Exp 6.4: Did the authors compare the GP-Huber model to other approaches?

---

> ### Comment · Reviewer_b8uR · 2024-11-26
>
> Since I have not received any reply from the authors, I would keep my scores.

---

> ### Author Response · Authors · 2024-11-26
>
> We thank the reviewers for their suggestions and feedback. Please find our comments:
>
> > The biggest advantage of the paper is that the advantage of the proposed approach is unclear.
>
> The proposed model addresses both extreme and near outliers in the covariate space and response by using the Huber density function as a likelihood (within GP framework) to limit their impact on the posterior of the latent parameter $f$. The unimodality and bounded influence of GP-Huber are established in Theorems 1 and 2 (see details @ https://anonymous.4open.science/r/GpHuber-6A2D/rebuttal.pdf). To the best of our knowledge, this is the first model to achieve this.
>
>
>
> > First, while the use of Huber likelihood is natural, there seems be a lack of discussions on the theoretical properties of Huber likelihood when compared to other robust likelihoods, e.g. student t, and their implications in Gaussian processes framework. Making this point clear would make the paper's contribution more significant other than just combining Gaussian process with any existing robust likelihood model.
>
> We will add this discussion in the revised paper. Specifically, we will add Theorem 2 to prove the robustness of the proposed approach.
>
> > On the empirical side, more thorough experimental details need to be presented. For each of the table presented, how many random simulations are run and what are the standard errors? See more comments on empirics in the questions below.
>
> We initially did not conduct random experiment. However, in the comparison presented in the Tables 1,2,3, and 4 @ https://anonymous.4open.science/r/GpHuber-6A2D/rebuttal.pdf, we conducted 10 random simulations and report the standard deviations in parenthesis.
>
> > Finally, it would be great of have a -brief introduction of the RCGP approach, which is served as an important baseline in the paper.
>
> We will add this in the revised version of the paper.
>
> > In the motivating figure 1, what is the result from your proposed approach? Also can you plot the true sinc function?
>
> Yes, we will share the figure shortly.
>
> > the paper claims that no additional hyperparameters need to be chosen. But there are indeed threshold parameters and contamination parameters required for the Huber model. The last paragraph in section 4 discusses one particular configuration, is this what the authors meant by eliminating the need for setting them?
>
> We mean exactly that. We will correct this in our revised paper.
>
> > Experiment 6.1, can you elaborate the difference between Case 1 v.s. Case 3 (or Case 4)? Is is just that in case 3 & 4, outliers in y space are closer to the main cluster of the data?
>
> In case 1, outliers in response $y^{(l)}$ are added (in focused or asymmetric pattern) at least 3 standard deviations away from the training data. In the case 3, $y^{(c)}$ were added within the 2 standard deviations. For case 4, outliers in covariates $x^{(l)}$ were added in  addition to the $y^{c}$. The locations to add outliers were chosen arbitrarily.
>
>
> > In table 1 (and other tables), no standard errors are reported. Also, in the first row of Tbale 1, Huber LA with RMSE 0.25 is better than Huber MCMC with RMSE 0.37, but why the latter is bolded. Same question regarding the first MAE row. Please go through the table to make sure the bolded numbers are indeed the best performance.
>
> Yes, we have corrected the bold values corresponding to the row of table 1 and the Tables with standard errors are presented in the rebuttal.pdf. Additionally, we have added pursuit weighting to all the baselines to the cases 1 and 3, where we added outliers in the input dimensions $x^{(l)}$.
>
> > Figure 2: where is the result for HuberMCMC? Also it would be nice to have a more concrete description on each method's qualitative performance.
>
> We will add the performance of HuberMCMC in the figure 2 in the revised paper.
>
> > Figure 3.a: How are the mean function and GP-Huber curves obtained? Figure 3.b: Describe the red bands and the dots.
>
> The mean function $T(t,\phi)$ is plotted by solving quadratic limb darkening function (an analytical function) which is the function of time $t$ and optical state parameters in $\phi$, indicated by red dots. The blue dots indicate flux measurements. GP-Huber (blue curve) plots its fit $f(t,X)$.
>
> > Exp 6.3: Did the authors compare the GP-Huber model to other approaches?
>
> We compared GP-Huber with the state-of-the-art method in the filed of transmission spectroscopy proposed by Gibson et.al., which uses GP to model instrumental systematics (please see the response to reviewer rnXe).  Modifying the experiment for Student's-t likelihood and Laplace was not a very straightforward task. For a real world dataset, we provided comparison of GP-Huber on Twitter flash crash dataset with RCGP and other baselines.

---

> > ### Comment · Reviewer_b8uR · 2024-11-27
> >
> > I would like to thank the authors for their thorough rebuttal.
> >
> > Followup question regarding the results: I do not see bolded values in tables. My observations are that I do not see a dominant performance by Huber MCMC or Huber Laplace. Would you agree with that? If not, could you highlight the conclusions from the updated table?

---

> ### Author Response · Authors · 2024-11-29
>
> We highlighted the values in the tables in \texttt{rebuttal.pdf}. HuberMCMC and HuberLA clearly outperform the baselines as expected. In outlier scenarios with $x^{(l)}$ and $y^{(l)}$, HuberMCMC performs significantly better than the pursuit-weighted baselines. Similarly, in cases with $y^{(c)}$, HuberLA shows better performance compared to the baselines. We notice that pursuit weighting improves the performance of Laplace and Gaussian likelihood GP models, while Student's-t likelihood and RCGP show minimal improvements. We hope this addresses your concerns and that you consider raising your scores.

---

> ### Author Response · Authors · 2024-12-01
>
> We hope that presented additional clarifications will allow you to recommend our work for full acceptance.

---

### Note · Authors · 2025-01-15

I have read and agree with the venue's withdrawal policy on behalf of myself and my co-authors.